# External evaluation of the Dynamic Criticality Index: A machine learning model to predict future need for ICU care in hospitalized pediatric patients

Anita K. Patel [1,2]*, Eduardo Trujillo-Rivera[1,2,3], James M. Chamberlain[2,4], Hiroki Morizono[3,5], Murray M. Pollack[1,2]

1 Department of Pediatrics, Division of Critical Care Medicine, Children's National Health System, Washington, DC, United States of America, 2 George Washington University School of Medicine and Health Sciences, Washington, DC, United States of America, 3 Children's National Research Institute, Children's National Hospital, Washington, DC, United States of America, 4 Department of Pediatrics, Division of Emergency Medicine, Children's National Hospital, Washington, DC, United States of America, 5 Department of Genomics and Precision Medicine, George Washington University School of Medicine and Health Sciences, Washington, DC, United States of America

* apatel4@childrensnational.org

**Data Availability Statement:** The minimal dataset is available the following link: https://zenodo.org/records/10436695.

## Abstract

### Objective

To assess the single site performance of the Dynamic Criticality Index (CI-D) models developed from a multi-institutional database to predict future care. Secondarily, to assess future care-location predictions in a single institution when CI-D models are re-developed using single-site data with identical variables and modeling methods. Four CI-D models were assessed for predicting care locations >6–12 hours, >12–18 hours, >18–24 hours, and >24–30 hours in the future.

### Design

Prognostic study comparing multi-institutional CI-D models' performance in a single-site electronic health record dataset to an institution-specific CI-D model developed using identical variables and modelling methods. The institution did not participate in the multi-institutional dataset.

### Participants

All pediatric inpatients admitted from January 1st 2018 –February 29th 2020 through the emergency department.

### Main outcome(s) and measure(s)

The main outcome was inpatient care in routine or ICU care locations.

**Funding:** Funding was provided by the Eunice Kennedy Shriver National Institute of Child Health and Human Development K23 Award number HD105978-01. (AKP) Its contents are solely the responsibility of the authors and do not necessarily represent the official views of the National Institute for Child Health and Human Development or the National Institutes of Health. https://www.nichd.nih.gov/ The funder had no role in study design, data collection and analysis, decision to publish, or preparation of the manuscript.

**Competing interests:** The authors have declared that no competing interests exist.

## Results

A total of 29,037 pediatric hospital admissions were included, with 5,563 (19.2%) admitted directly to the ICU, 869 (3.0%) transferred from routine to ICU care, and 5,023 (17.3%) transferred from ICU to routine care. Patients had a median [IQR] age 68 months (15–157), 47.5% were female and 43.4% were black. The area under the receiver operating characteristic curve (AUROC) for the multi-institutional CI-D models applied to a single-site test dataset was 0.493–0.545 and area under the precision-recall curve (AUPRC) was 0.262–0.299. The single-site CI-D models applied to an independent single-site test dataset had an AUROC 0.906–0.944 and AUPRC range from 0.754–0.824. Accuracy at 0.95 sensitivity for those transferred from routine to ICU care was 72.6%-81.0%. Accuracy at 0.95 specificity was 58.2%-76.4% for patients who transferred from ICU to routine care.

## Conclusion and relevance

Models developed from multi-institutional datasets and intended for application to individual institutions should be assessed locally and may benefit from re-development with site-specific data prior to deployment.

## Introduction

Over two million children are hospitalized annually in the US [1], and approximately 15% require transfer from routine wards to intensive care units (ICUs) [2]. If the need for ICU transfer is not recognized expediently, patients may develop new morbidity and/or mortality [3]. While experienced clinicians are generally effective at determining patient severity, less experienced providers may inadequately assess and synthesize the clinical information. Simple severity warning scores, such as the Pediatric Early Warning Score (PEWS), may be sensitive for detecting emergent clinical deterioration but lack specificity and positive predictive value (precision), resulting in a high proportion of false positives and resultant alarm fatigue [4–7]. The addition of sensitive and specific severity assessments with high precision for hospitalized children could improve quality of care by detecting potential clinical deteriorations or improvements that might have been unappreciated, providing an opportunity for earlier therapeutic changes and improved outcomes.

Recently, we developed a set of four models developed from separate neural networks that predict future care needs assessed as care location using a new severity measure, the Dynamic Criticality Index (CI-D) [8–10]. The CI-D uses physiology, therapy, and therapeutic intensity variables to predict the need for ICU care [8–10]. The outcome of ICU care was used because it has direct clinical relevance and the traditional severity of illness outcome, mortality, is relatively rare in children. We demonstrated that the CI-D models were calibrated to receiving ICU care (or the converse, routine care) in the future time periods of >6–12 hours, >12–18 hours, > 18 hours to 24 hours, and >24–30 hours [10]. Four separate models were developed for prediction of care location in each of the four future time intervals. The four models developed from neural networks comprising the CI-D performed well in a prospective multi-institutional validation sample and a simulated children's hospital, and was not influenced by institutional characteristics [10].

It is important to evaluate the performance of models developed from multi-institutional databases in individual sites [11]. This is especially important when the prediction outcome,

such as care area, is influenced by institutional practices that may not be captured through modeling of physiology. Models developed from multi-institutional data could have loss of performance in individual sites or could be improved if they were recalibrated for individual sites. We performed this study to 1) assess the performance of the CI-D models developed from a multi-institutional database to predict future care locations when applied to a single site that did not participate in the multi-institutional dataset [10], and 2) to apply the identical variables and modeling methods to a single-site dataset to develop models optimized for the single site, and compare this performance to the multi-institutional CI-D models' performances.

## Methods

### Setting

The study was performed at Children's National Hospital, an academic hospital with 323 beds inclusive of 48 bed Pediatric ICU beds and 24 Cardiac ICU beds. The patient sample included all inpatient admissions directed or coordinated through the Emergency Department to the inpatient service from January 1, 2018 to February 29, 2020. The end date was chosen to reflect care prior to the novel coronavirus pandemic. Exclusions included patients over 21 years of age and those admitted to the neonatal ICU. This study was approved by the Institutional Review Board at Children's National Health System.

### Dataset

The dataset was extracted from the electronic health record (EHR) and de-identified. Date/time fields were offset to maintain relative time differences. The outcome of care location (routine care or ICU care) was defined by admission and/or transfer orders. All independent variables from the CI-D were included, representing physiology, treatment, and care intensity [8, 10]. This included age, 30 laboratory tests, 6 vital signs, and 1113 medications classified into 143 categories [8, 10]. Medication data were extracted from the medication administration record using start and end times. Medication classes were categorized using the Multum™ system [12]. A comprehensive list of independent variables and details of medication classification are reported in S1 and S2 Files. Other data included demographic and diagnostic information categorized by the International Classification of Diseases 10th Edition (ICD-10) [13]. Diagnoses were used for descriptive purposes, but not for modelling, because they were determined at discharge. No new variables were included in the single-site modeling.

### Application of the multi-institutional model to the single-site dataset

The performances of the multi-institutional CI-D models applied to the full single-site dataset were evaluated for the four future time predictors. The institution did not participate in the multi-institutional dataset. Model discrimination was assessed by the area under the receiver operating characteristic curve (AUROC) and the area under the precision recall curve (AUPRC) [14]. The classification performance of the model was assessed by the following metrics at a decision threshold of 0.5: sensitivity, precision (positive predictive value), accuracy, specificity, negative predictive value, and F1 score. Sensitivity (true positive) indicates the proportion of correctly identified patients transferred to, or remaining in, the ICU for the outcome time period while specificity (true negative) indicates the proportion of correctly identified patients remaining in a routine care area or transferred out of the ICU to a routine care area for the outcome time period. The number needed to evaluate is calculated as 1/precision. Calibration was assessed using the differences between the observed and expected

proportions of ICU outcomes over the full range of risk. The number of calibration intervals for each model was >240 with a minimum of 200 patients in each risk interval. We report the percentage of intervals with no statistical evidence for difference [15, 16]. We computed the regression line for comparison to the ideal using the calibrated model's output for observed and expected proportions of ICU care. Optimal calibration plot performance includes intercept = 0, slope = 1, $R^2$ = 1, and ≤5.0% of risk intervals with a statistically difference (p < 0.05) between the observed and expected proportions and Cohen's h <0.2 [17].

Performance of the multi-institutional CI-D applied to the single-site test dataset was compared to the multi-institutional CI-D models' performances applied to a previously reported simulated children's hospital [10], consisting of all 7,054 pediatric admissions in the Health Facts® database from January 2016 to June 2018 with an ICU prevalence rate of 20.0%, similar to the single site sample of 22.5% [10]. Performance data for the four multi-institutional models applied to the multi-institutional dataset were: AUROC = 0.946–0.948, AUPRC = 0.726–0.742, sensitivity = 0.545–0.590, precision (positive predictive value) = 0.748–0.771, accuracy = 0.921–0.925, specificity = 0.972–0.977, negative predictive value = 0.937–0.994, and F1 score = 0.639–0.660. The models were well calibrated, with $R^2$'s ranging from 0.979–0.982, slopes 0.941–0.943, intercepts 0.016 –-0.013, and Cohen's h < 0.2.

## Machine learning methodology

The machine learning methodology for the single-site CI-D models from the single-site data set followed the identical process as that of the multi-institutional models with identical variables and congruent model architecture [8, 10]. The hospital course was discretized into consecutive 6-hour time periods. Six hours was selected because data acquisition for non-ICU care patients is relatively infrequent compared with ICU patients. A 6-hour time periods allowed for sufficient time to obtain a set of vital signs, and frequently a medication and/or lab result to inform future care location predictions. Each time period was categorized into ICU or routine care; we excluded time periods when patients were transitioning between routine and ICU care. We randomly selected 75% of patients for training, 13% for validation, and 12% for testing. The training set was used for model development, the validation set was used to fine-tune parameters and prevent over-fitting, and the test set was used to evaluate model performance. We report performance metrics on the test set.

Independent neural networks calibrated to risk of ICU care were developed for four future time periods (>6–12 hours, >12–18 hours, >18–24 hours, and >24–30 hours) to optimize performance for each time period. Therefore, prediction of ICU or non-ICU care were based on a single model for each time period. The four future time periods were chosen as clinically relevant time periods for both immediate and future resource allocation planning for potential increases and decreases in patient care needs. Inputs included variables in the present time period, and up to 24 hours in preceding time periods. New future predictions in a 6-hour time period were made only if there were at least one new data element. Missing data were imputed since our machine learning methods required laboratory and vital sign inputs in each time period. The imputation methodology has been reported [8, 18] and is detailed in S3 File. The single-site machine learning models used identical data elements and modeling methodology as the multi-institutional CI-D models [8, 10]. A single neural network for binary classification for ICU outcome was developed for each future prediction time-period. The raw output of the neural networks was a continuous number between 0 and 1 which was converted to a binary classifier (ICU care or non-ICU care) by choosing a cut point. A neural network output below the set cut point was classified as non-ICU care, whereas an output above the set cut point was classified as ICU care. The objective of each training epoch was to minimize the training

binary cross entropy while monitoring local maximization of the validation Mathew correlation coefficient (MCC) at a cut point of 0.5 while not allowing for drift from the training MCC at same cut point. We started with a neural network with a single hidden layer, and logit output and consecutively increased the number of nodes while monitoring for overfitting by computing the MCC, sensitivity, specificity, precision, negative predictive value at cut points of 0.15, 0.5, and 0.9 for binary classification in the training and validation sets. If overfitting was found, we add L2 regularization, and layer node dropouts with parameters tuned to maintain similar metrices on the validation and training sets. We increased the number of nodes if there was not overfitting or if it was eliminated using the outlined procedure. When there were no additional gains on the performance metrics, we added another hidden layer and repeated the process, increasing the number of hidden layers until our regularization attempts were unsuccessful in avoiding overfitting. We keep the best model as measured by the Mathew's correlation coefficient without overfitting. Glorot uniform was used to initialize each layer's weight, and optimized using RMSprop. An initial learning rate of 0.0002 was used with minibatches each consisting of a size of 10,000 data points [19]. The final neural networks were fully connected with five sequential hidden layers, but different number of nodes across hidden layers and models. Each layer had $L_1$, $L_2$ and dropout node regularizations with varying smoothing parameters. Each model has an output layer with one node and logistic activation. Each neural network was calibrated to the future risk of ICU care using B-splines polynomials and logistic regression.

The single-site CI-D models' classification performance was first assessed in the test sample using the same performance metrics detailed above. Second, we assessed the performance of the models using sensitivities of 0.85, 0.90, 0.95 and 0.99 because these represent potential thresholds for clinical decision making. Third, we assessed accuracy in predicting a change in care location exclusively in patients who transitioned from routine to ICU care and/or ICU to routine care. For example, a patient who transferred from routine to ICU care could have a prediction from the >24–30 hour model, >18–24 hour model, >12–18 hour model, and >6–12 hour model if they had new data available in each time period and were in the routine care location for at least 30 hours prior to transfer, whereas a patient who only had 12 hours in a routine care area prior to transfer to the ICU would have a prediction from the >6–12 hour model only. Modeling and statistical analysis were performed by a single author (EATR).

We explored the variable importance using a Local Interpretable Model-Agnostic Explanation (LIME) approach (LIME R package) to support the possibility that practice differences including ICU admission criteria could, in part, account for the performance of the multi-institutional models in the single-site [20]. Briefly, the LIME approach is based on the assumption that every model performs like a linear prediction model for each prediction and the hierarchy of covariate importance is preserved. The collection of individual linear models provides an interpretation of the covariate importance in the final model. We ranked the most important covariates across all ICU risk predictions by computing the percentage of times each variable was among the 30 most important covariates. We evaluated the four single-site machine learning models on the test single-site dataset and the four multi-institutional models on the test multi-institutional dataset. We selected the subset of cases where the predicted risk of need for ICU care was between 0.245 and 0.255 (referred to as "risk condition"). We selected a standard risk condition for the purpose of variable importance comparison between models among patients with a similar risk of ICU admission. Using this risk condition, in the single-site test dataset, a total of 201–202 cases were assessed per prediction model, and between 953 and 2,328 cases were assessed from the multi-institutional test set. We focused on covariates that might indicate ICU practices differences and used the following as indicators of practice differences: medications and coma scores. Medications were focused on because care practices

at some institutions favor the use of specific medications in their ICU patients that are rarely administered on the inpatient unit. Coma score was identified because the ability to care for children with altered states of consciousness in non-ICU areas varies among institutions.

## Results

There were 29,037 inpatient admissions (Table 1), of which 5,563 (19.2%) were admitted directly to the ICU, 869 (3.0%) were transferred from routine to ICU care, and the remainder were admitted to routine care until discharge. A total of 5,023 of the ICU admissions (78.1% of ICU admissions) were transferred to routine care, and the remainder were discharged from the hospital directly from the ICU. Median age was 68 months (IQR 15–157), 47.5% were female, 43.3% were black, and median hospital length of stay (LOS) was 2.7 days (IQR 1.6–

**Table 1. Population characteristics of children's national patient sample.**

| Characteristic | All | ICU | Routine Care Only | p-value[c] | Transfers Routine to ICU | Transfers ICU to Routine |
|---|---|---|---|---|---|---|
| Admissions (n (%)) | 26,401 | 6,432 (22.5) | 19,969 (28.8) | <0.001 | 869 (3.3) | 5,023 (19.0) |
| 6-hour Time periods (n (%)) | 529,538 | 110,136 (20.8) | 419,402 (79.2) | <0.001 | 5,458 (1.0) | 4,092 (0.8) |
| Age (Months)[a] | 68.0 (15.0–157.0) | 49.0 (12.0–49.0) | 208.0 (198.0–223.0) | <0.001 | 48.0 (10.0–140.0) | 62.0 (23.0–149.0) |
| Female (n (%)) | 13,785 (52.2) | 3,254 (50.6) | 9,705 (48.6) | <0.001 | 399 (45.9) | 2,278 (45.4) |
| Black | 12,564 (47.6) | 2,909 (45.2) | 9,655 (48.3) | <0.001 | 399 (45.9) | 2,302 (45.8) |
| White | 5,240 (19.8) | 1,315 (20.4) | 3,925 (19.7) | <0.001 | 170 (19.6) | 1,004 (20.0) |
| Other-Unknown | 8,597 (32.6) | 2,208 (34.3) | 6,389 (32.0) | 0.598 | 300 (34.5) | 1,717 (34.2) |
| Hospital LOS (days)[a] | 2.4 (1.3–4.7) | 3.7 (2.2–7.7) | 1.9 (1.0–3.9) | <0.001 | 7.0 (3.9–14.7) | 3.7 (2.3–7.2) |
| ICU LOS (days)[a] | | 2.0 (1.1–3.8) | | <0.001 | 2.8 (1.5–5.6) | 1.8 (1.0–3.5) |
| Diagnostic Group[b] (n (%)) | | | | | | |
| Respiratory | 11,580 (45.5) | 4,499 (72.8) | 7,081 (36.8) | <0.001 | 519 (62.9) | 3,536 (70.4) |
| Endocrine, Nutritional, Metabolic, and Immune | 6152 (24.2) | 1490 (24.1) | 4662 (24.2) | 0.920 | 271 (32.9) | 1166 (23.2) |
| Gastrointestinal | 6284 (24.7) | 1106 (17.9) | 5178 (26.9) | <0.001 | 228 (27.7) | 802 (16.0) |
| Infectious | 5415 (21.3) | 1950 (31.6) | 3465 (18.0) | <0.001 | 234 (28.3) | 1550 (30.9) |
| Injury and poisoning | 3821 (15.0) | 866 (14.0) | 139 (16.8) | 0.072 | 672 (77.3) | 2955 (58.8) |
| Neurological | 5107 (20.1) | 1543 (25.0) | 3564 (18.5) | <0.001 | 225 (27.3) | 1021 (20.3) |
| Neoplasms | 885 (3.5) | 186 (3.0) | 699 (3.6) | 0.400 | 52 (6.3) | 159 (3.2) |
| Hematologic | 4589 (18.0) | 693 (11.2) | 3896 (20.2) | <0.001 | 157 (19.0) | 569 (3.2) |
| Cardiovascular | 5555 (21.8) | 1691 (27.4) | 3864 (20.1) | <0.001 | 303 (36.7) | 303 (6.0) |
| Musculoskeletal | 1932 (7.6) | 373 (6.0) | 1559 (8.1) | <0.004 | 73 (8.8) | 251 (5.0) |
| Dermatologic | 2616 (10.3) | 467 (7.6) | 2149 (11.2) | <0.001 | 370 (7.7) | 2149 (42.8) |
| Genitourinary | 2164 (8.5) | 408 (6.6) | 1756 (9.1) | <0.001 | 80 (9.7) | 310 (6.2) |
| Ophthalmologic | 981 (3.9) | 321 (5.2) | 660 (3.4) | 0.015 | 57 (6.9) | 231 (4.6) |
| Congenital | 2914 (11.5) | 1020 (16.5) | 1894 (9.8) | <0.001 | 161 (19.5) | 717 (14.3) |
| Not Otherwise Specified | 12310 (48.4) | 2774 (44.9) | 9536 (49.5) | <0.001 | 387 (46.9) | 2019 (40.2) |

The patients included in the four models for training, validation, and testing. Patients in the two transfer groups (routine to ICU and ICU to routine) were not mutually exclusive; therefore, they were not compared statistically.

Abbreviations: LOS = Length of Stay

a Median (25th percentile, 75th percentile)

b Diagnostic categories were used for descriptive purposes only. A single encounter could have diagnoses in multiple groups.

c Patients who were admitted to the ICU at any time during their hospital admission were compared to patients who were never admitted to the ICU (Routine Care Only) for significance via univariate analysis.

5.1). Compared to ICU admissions, routine care admissions were older (median 208 vs. 49 months, p <0.001), and had shorter hospital LOS (median 1.9 vs. 3.7 days, p <0.001). Most diagnostic categories differed between ICU and routine care admissions (p < 0.01). Overall, there were 110,136 (20.8%) ICU time periods and 419,402 (79.2%) routine care time periods.

The performance metrics of the multi-institutional CI-D models applied to the single-site test dataset were predominantly poor. For the four future prediction time periods, the AUROC = 0.493–0.545 (Fig 1A), and the AUPRC = 0.259–0.299 (Fig 1B). Performance metrics for the 4 models at a decision threshold of 0.5 on the test set (Table 2A) were: sensitivity 0.007–0.008, precision 0.378–0.628, accuracy 0.790–0.794, specificity 0.997–0.999, negative predictive value 0.792–0.794, and F1 score 0.013–0.016. Calibration plots (Fig 1C) revealed slopes = 0.344 to 0.406, intercepts = -0.072 to 0.065, and $R^2$'s = 0.491 to 0.647.

The performance metrics of the single-site CI-D models applied to the single-site test sample (Fig 2, Table 2B) were substantially better than the performance of the multi-institutional model. For the four future prediction time periods, the AUROC = 0.906–0.944 (Fig 2A) and the AUPRC = 0.754–0.824 (Fig 2B). Performance metrics for the 4 models at a decision threshold of 0.5 (Table 2B) were: sensitivity 0.584–0.718, precision 0.737–0.765, accuracy 0.871–0.896, specificity 0.943–0.946, negative predictive value 0.897–0.928, and F1 score 0.871–0.896. Calibration plots (Fig 2C) revealed that the single-site models were well calibrated with slopes = 1.04 to 1.06, intercepts = -0.004 to -0.013, and $R^2$'s $\geq$ 0.99.

An assessment of covariate importance comparing the single-site models and the multi-institutional models using the LIME approach is presented in Fig 3. Among the full list of covariates, 20%-30% of the 30 most frequent covariates used for ICU predictions made in the multi-institutional models were not among the 30 most frequent variables in the single-site models. Similarly, between 17%-25% of the most frequent covariates used for ICU predictions in the single-site models were not among the 30 most frequent covariates used in the multi-institutional models. Certain medications and coma score were more frequently in the 30 most frequent single-site covariates, consistent with unique practices patterns in this site. For example, the proportion of time on antipsychotics, bronchodilators, and number of lubricants/irrigations were among the 30 most frequently utilized covariates in the single-site model, but not in the multi-institutional model. Multiple nasal lubricants and anti-psychotics are medication categories commonly administered to invasively ventilated patients at the single-site and all children less than 3 years old requiring continuous bronchodilator therapy must be automatically admitted to the ICU, practices that may not be shared at other institutions but are unique to the single-site. Average coma score was among the 30 most frequently used covariates in the single-site model, but not in the multi-institutional model. Similarly, minimum and maximum coma score were more frequently among the 30 most frequent covariates used in the single-site model when compared to the multi-institutional model (3.4% in the single-site model vs 2.5% in the multi-institutional model, and 3.4% in single-site model vs 1.2% in multi-institutional model respectively). The single-site has a policy which requires all patients on every one-hour neurologic assessments to be monitored in the ICU; this practice is not shared at all institutions and may account for coma score values cared for in the single-site ICU to be different than in other institutions.

Table 3 shows the performance metrics of the single-site model applied to test patients for sensitivities of 0.85, 0.90, 0.95, and 0.99 to display alternative clinical decision thresholds. All models followed a similar pattern; for example, as sensitivity increased from 0.85 to 0.99 for the >6–12-hour time period, precision decreased from 0.663 to 0.370, accuracy decreased from 0.850 to 0.650, specificity decreased from 0.887 to 0.569, and negative predictive value increased from 0.958 to 0.995. There was also a decrement of performance as the prediction time period lengthened. For example, at a sensitivity of 0.95, accuracy decreased from 0.797 to

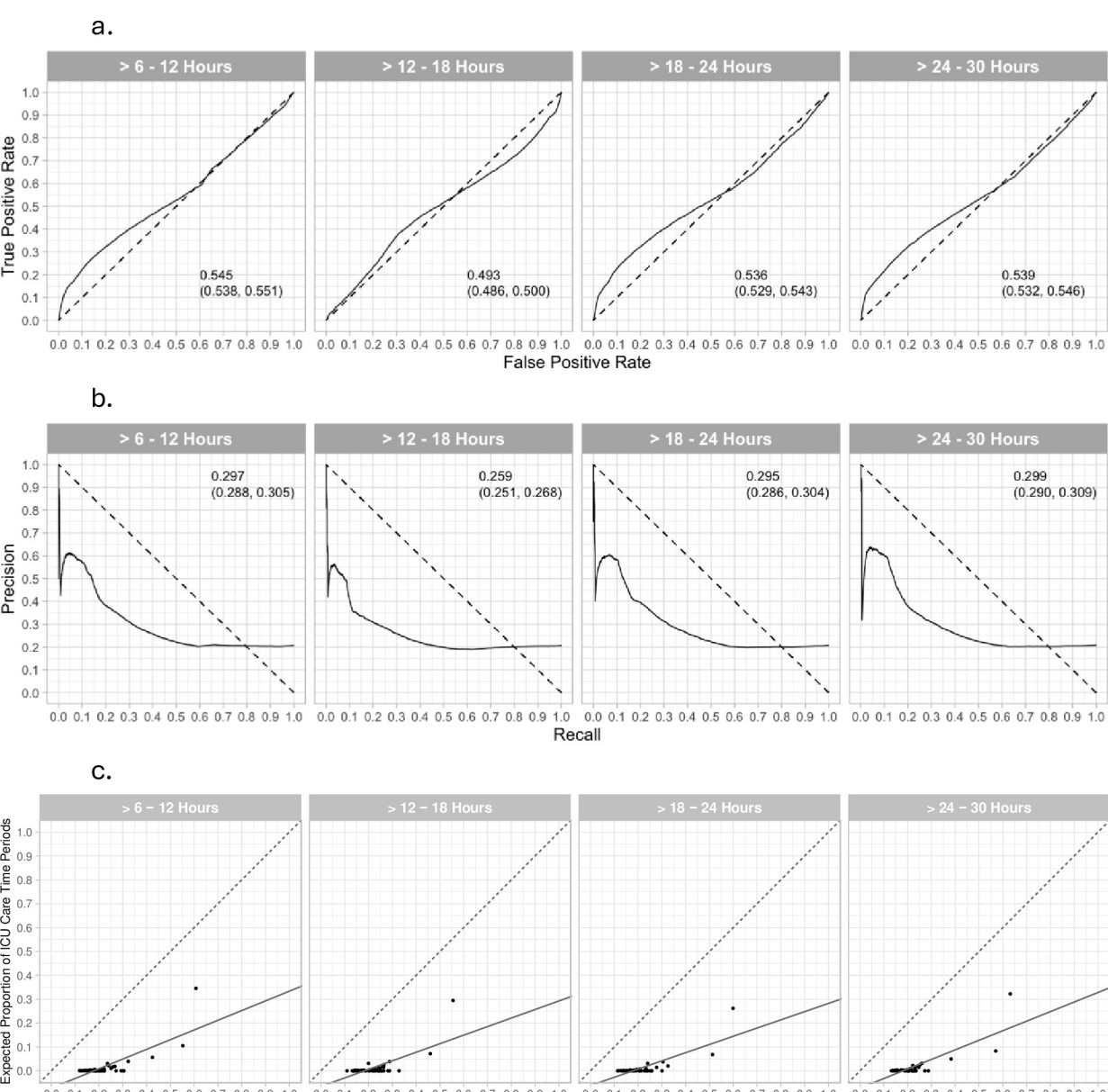

**Fig 1.** Area under the receiver operating characteristic (AUROC) curves (a), area under the precision recall (AUPRC) curves (b), and calibration plots (c) for the multi-institutional Dynamic Criticality Index models applied to the single-site test dataset (n = 26,401). Outcomes were ICU or routine care for the four future time periods of >6–12 hours, >12–18 hours, >18–24 hours, and >24–30 hours. The 95% CIs are included in panels a and b. **a. AUROC for four future time periods.** The area under the receiver operating characteristic curves for classifying care as non-ICU or ICU for the respective future time periods are presented in each panel. **b. AUPRC for four future time periods.** The area under the precision-recall curves and 95% CIs are included in each panel. The areas were computed with integral approximations, and the CIs were computed using a logit method [21]. **c. Calibration plots for four future time periods.** The y-axis shows the expected proportion of ICU care areas for the time periods based on the risk intervals, and the x-axis shows the observed proportion of ICU care areas in the time periods. A linear regression is reported in each panel, with their respective $R^2$, and the fitted mean is represented with the solid line. The line of identity is the dashed line. The risk intervals were composed by requiring a minimum of 200 data points per risk interval, resulting in 389, 375, 365, and 356 risk intervals in the four future time period models, respectively. The circles indicate the observed and the expected proportions of ICU 6-hour time periods over ascending Criticality Index intervals. 0.0%, 0.0%, 0.3%, and 0.0% of the risk intervals within each plot have a Cohen's h value <0.2 indicating large effect sizes for differences between observed and expected proportions.

**Table 2. Performance metrics of the Dynamic Criticality Index models developed from the multi-institutional database applied to the single-site test dataset (A) and the single-site Dynamic Criticality Index models applied to the single-site test dataset (B).**

| Prediction Time Period | Sensitivity (1) | Precision (2) | Accuracy (3) | Specificity (4) | Negative Predictive Value (6) | F1 score (6) |
|---|---|---|---|---|---|---|
| A. Multi-Institutional Dynamic Criticality Index | | | | | | |
| >6–12 Hours | 0.008 (0.007, 0.010) | 0.428 (0.365, 0.494) | 0.793 (0.789, 0.796) | 0.997 (0.997, 0.998) | 0.794 (0.791, 0.798) | 0.016 (0.014, 0.019) |
| >12–18 Hours | 0.007 (0.005, 0.008) | 0.628 (0.536, 0.712) | 0.794 (0.790, 0.797) | 0.999 (0.999, 0.999) | 0.794 (0.791, 0.798) | 0.013 (0.011, 0.015) |
| >18–24 Hours | 0.007 (0.006, 0.009) | 0.567 (0.480, 0.650) | 0.793 (0.789, 0.797) | 0.999 (0.998, 0.999) | 0.794 (0.790, 0.797) | 0.014 (0.012, 0.016) |
| >24–30 Hours | 0.007 (0.006, 0.009) | 0.378 (0.312, 0.450) | 0.790 (0.787, 0.794) | 0.997 (0.996, 0.997) | 0.792 (0.788, 0.796) | 0.014 (0.012, 0.017) |
| B. Single-site Dynamic Criticality Index Model | | | | | | |
| >6–12 Hours | 0.718 (0.710, 0.726) | 0.765 (0.757, 0.773) | 0.896 (0.894, 0.899) | 0.943 (0.940, 0.945) | 0.928 (0.925, 0.930) | 0.896 (0.894, 0.899) |
| >12–18 Hours | 0.656 (0.647, 0.665) | 0.750 (0.741, 0.758) | 0.884 (0.881, 0.886) | 0.943 (0.941, 0.945) | 0.913 (0.911, 0.916) | 0.884 (0.881, 0.886) |
| >18–24 Hours | 0.608 (0.599, 0.618) | 0.748 (0.739, 0.757) | 0.877 (0.874, 0.880) | 0.947 (0.944, 0.949) | 0.903 (0.900, 0.906) | 0.877 (0.874, 0.880) |
| >24–30 Hours | 0.584 (0.575, 0.594) | 0.737 (0.727, 0.747) | 0.871 (0.868, 0.874) | 0.946 (0.943, 0.948) | 0.897 (0.894, 0.900) | 0.871 (0.868, 0.874) |

1. Sensitivity = true positive rate = proportion correctly identified as cared for in the ICU.

2. Precision = Positive predictive value = true positives/[true positive + false positive]. Number needed to evaluate = 1/precision.

3. Accuracy = (true positives + true negatives)/(positives + negatives)

4. Specificity = true negatives = cared for in routine care.

5. Negative predictive value = true negatives/[true negatives + false negatives]

6. The F$_1$ score is a measure of accuracy with a maximum score of 1. It is the harmonic mean of precision and sensitivity.

Performance metrics are computed at a decision threshold of 0.5. Identification of true positives is most relevant to identifying those patients expected to transfer to the ICU from routine care areas or remain in the ICU. The identification of true negatives is most relevant to identifying those patients expected to transfer to routine care from ICU care areas or remain in routine care areas. The data shown are the estimates and 95% confidence intervals (CI).

0.673 from the >6–12-hour future time period to >24 to 30-hour future time period, precision decreased from 0.505 to 0.383, specificity decreased from 0.757 to 0.600, and the negative predictive value decreased from 0.983 to 0.979. The number needed to evaluate to detect a routine care patient needing ICU care increased from 2.0 to 2.6.

Fig 4 shows the accuracy of all four models in predicting a change in care location for the 124 patients (4.1% of the test sample) who transferred from routine to ICU care and the 478 patients (15.8% of the test sample) who transferred from ICU to routine care. Accuracy is presented at 95% sensitivity. Similar data for sensitivities and specificities of 0.85, 0.9, and 0.99 are shown in S4 File. The correct predictions for transfer from routine to ICU care ranged from 72.6% -81.0% across all four models. In general, the accuracy remained relatively stable over the 4 models for each sensitivity, improving as the sensitivity increased. The accuracy for transfer from ICU to routine care at a specificity of 0.95 ranged from 58.2% - 76.4% for the 4 time periods. Similar to the analysis of different sensitivities, the accuracy remained relatively stable within each specificity, with a trend towards improvement as the time period increased and as the specificity increased.

## Discussion

We assessed the performance of a set of machine learning models developed in a multi-institutional database to predict ICU or routine care at four future time periods in a single site [10].

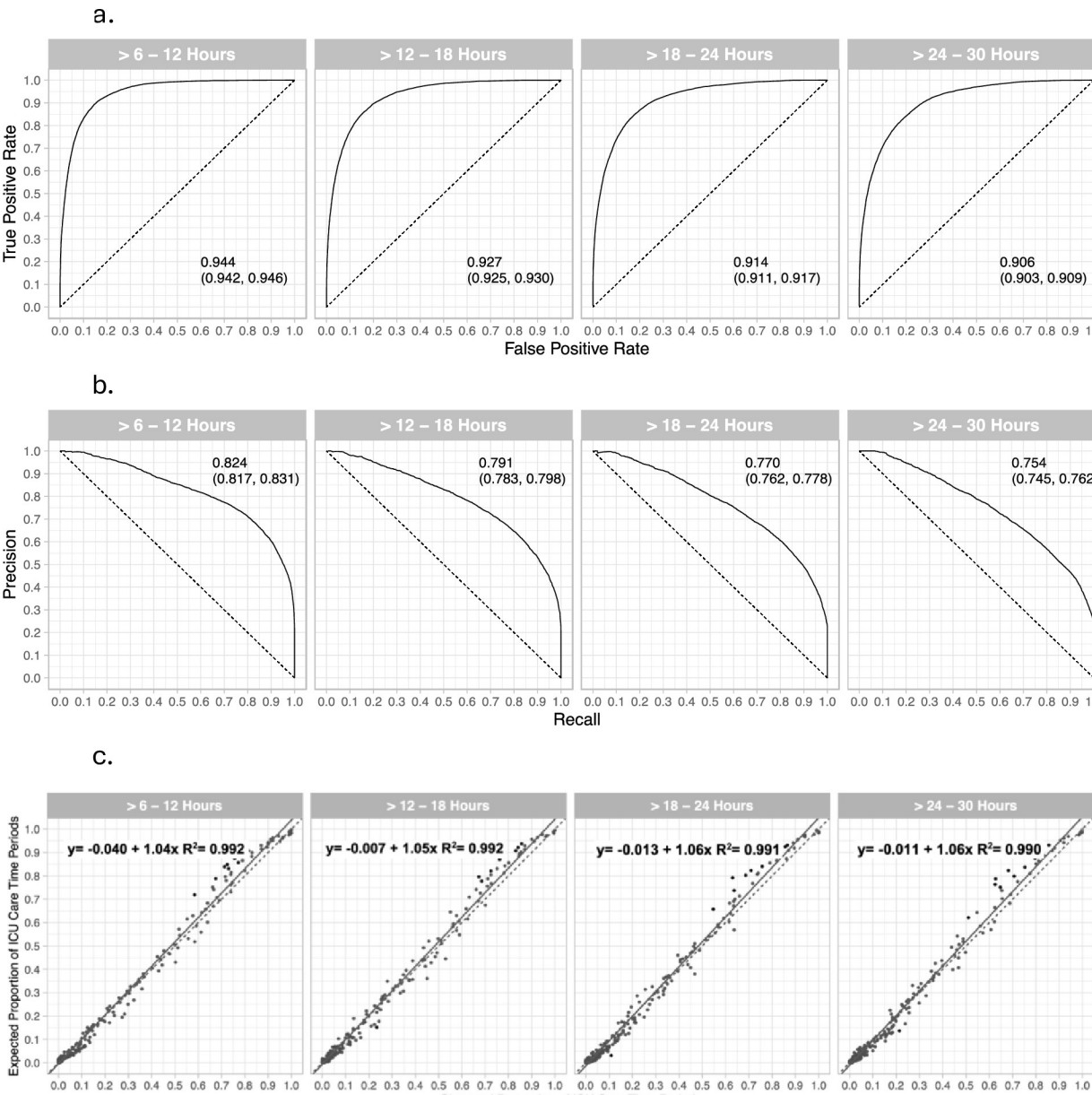

**Fig 2.** Area under the receiver operating characteristic (AUROC) curves (a), area under the precision recall (AUPRC) curves (b), and calibration plots (c) for the single-site Dynamic Criticality Index models applied to the single-site test dataset (n = 3,018). Outcomes were ICU or routine care for the four future time periods of >6–12 hours, >12–18 hours, >18–24 hours, and >24–30 hours. The 95% CIs are included in panels a and b. **a. AUROC for four future time periods.** The area under the receiver operating characteristic curves for classifying care as non-ICU or ICU for the respective future time periods are presented in each panel. **b. AUPRC for four future time periods.** The area under the precision-recall curves and 95% CIs are included in each panel. The areas were computed with integral approximations, and the CIs were computed using a logit method [21]. **c. Calibration plots for four future time periods.** The y-axis shows the expected proportion of ICU care areas for the time periods based on the risk intervals, and the x-axis shows the observed proportion of ICU care areas in the time periods. A linear regression is reported in each panel, with their respective $R^2$, and the fitted mean is represented with the solid line. The line of identity is the dashed line. The risk intervals were composed by requiring a minimum of 200 data points per risk interval, resulting in 287, 272, 250, and 242 risk intervals in the four future time period models, respectively. Within each interval, we computed the average expected risk of ICU admission and the observed risk of ICU admission. The circles indicate the observed and the expected proportions of ICU 6-hour time periods over ascending Criticality Index intervals. 96.9%, 97.1%, 96.0%, and 95.0% of the risk intervals within each plot have a Cohen's h value <0.2, indicating there are small effect size differences between observed and expected proportions.

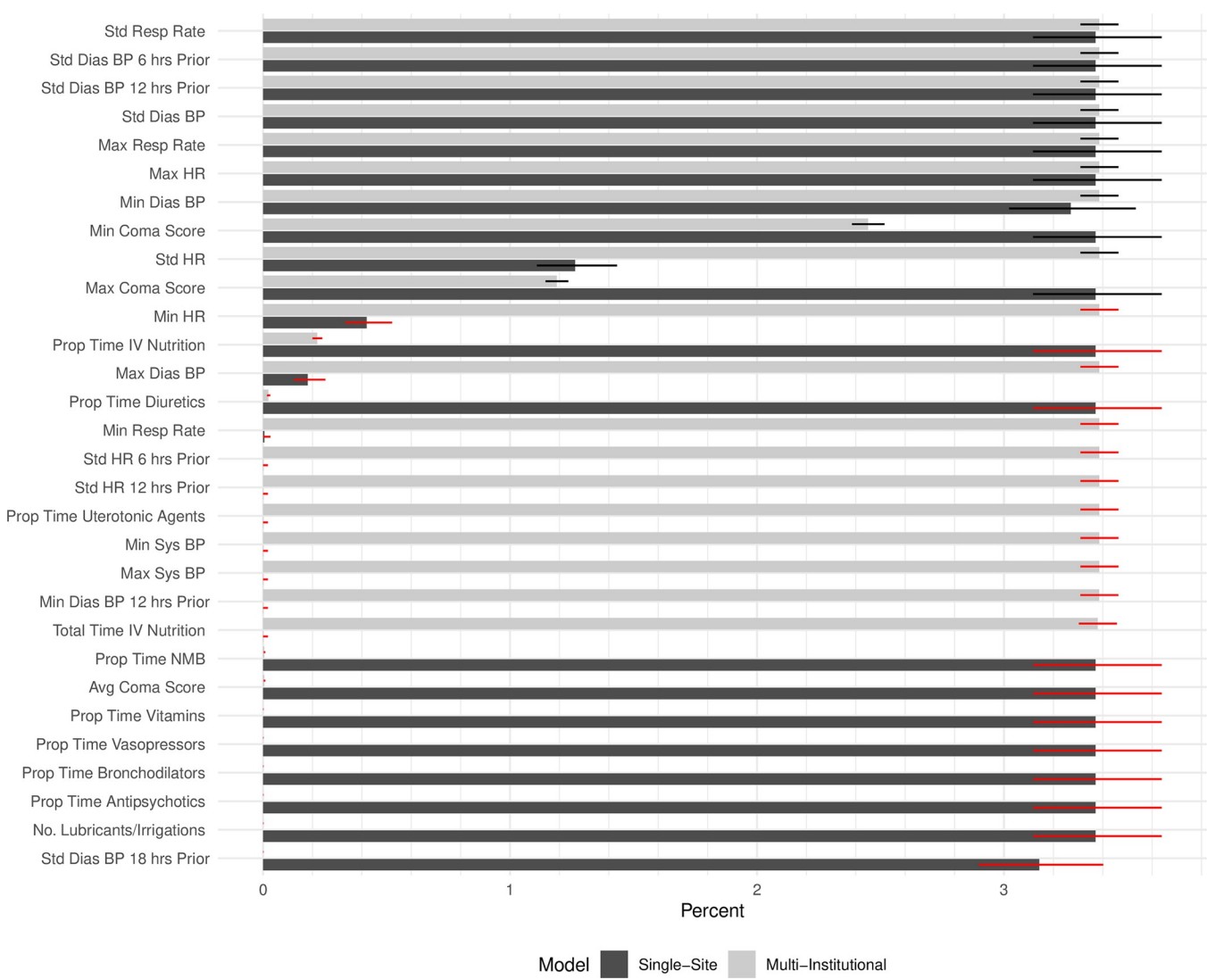

**Fig 3. Variable importance for the single-site and multi-institutional Dynamic Criticality Index models.** Percentages of the 30 most frequently important covariates across all models in the single-site (light grey) and multi-institutional (dark grey) models are presented as determined by the LIME methodology. The red and black lines represent the 95% confidence interval. Red confidence intervals indicate that the single-site and multi-institutional confidence intervals do not overlap and the pairs of features have a Cohen's h >0.2, indicating that the proportion of use has a practical difference. The data reported was computed using the cases where the predicted risk is between 0.245 and 0.255.

Despite excellent performance in an independent multi-institutional dataset [10], the multi-institutional models had substantially reduced performance in the single-site dataset. We then applied the identical methodology with identical data elements and developed institution-specific models. The single-site models had excellent performance metrics in a range appropriate for future clinical testing. These results are consistent with, but more extreme, than the results of Brajer et al., who found that their multi-institutional model estimating mortality risk benefited from local re-training, re-calibration, and validation to improve local performance [11]. The more extreme deviation of our local results from the multi-institutional model performance is likely the result of our models' outcome, future care location, which is dependent on local factors such as the number and availability of beds in ICU, nursing capacity, physician availability and training, and local care protocols. This possibility was supported by the LIME

**Table 3. Single-site Dynamic Criticality Index performance metrics for sensitivities of 0.85, 0.90, 0.95, and 0.99 representing four potential decision thresholds for the four future time periods.**

| Prediction Time Period | Sensitivity (1) | Precision (2) | Accuracy (3) | Specificity (4) | Negative Predictive Value (5) | F1 score (6) |
|---|---|---|---|---|---|---|
| >6–12 Hours | 0.850 (0.844, 0.856) | 0.663 (0.655, 0.670) | 0.880 (0.877, 0.882) | 0.887 (0.884, 0.890) | 0.958 (0.956, 0.960) | 0.745 (0.740, 0.750) |
| >6–12 Hours | 0.900 (0.895, 0.905) | 0.603 (0.596, 0.610) | 0.857 (0.854, 0.860) | 0.846 (0.842, 0.849) | 0.970 (0.968, 0.972) | 0.722 (0.717, 0.727) |
| >6–12 Hours | 0.950 (0.946, 0.954) | 0.505 (0.498, 0.511) | 0.797 (0.794, 0.800) | 0.757 (0.753, 0.761) | 0.983 (0.982, 0.984) | 0.659 (0.654, 0.664) |
| >6–12 Hours | 0.990 (0.988, 0.992) | 0.370 (0.365, 0.375) | 0.650 (0.646, 0.653) | 0.561 (0.556, 0.565) | 0.995 (0.994, 0.996) | 0.539 (0.534, 0.543) |
| >12–18 Hours | 0.850 (0.843, 0.856) | 0.595 (0.587, 0.602) | 0.850 (0.846, 0.852) | 0.849 (0.846, 0.853) | 0.956 (0.954, 0.958) | 0.700 (0.694, 0.705) |
| >12–18 Hours | 0.900 (0.894, 0.905) | 0.532 (0.525, 0.539) | 0.816 (0.812, 0.819) | 0.794 (0.790, 0.797) | 0.968 (0.966, 0.970) | 0.668 (0.663, 0.674) |
| >12–18 Hours | 0.950 (0.946, 0.954) | 0.443 (0.436, 0.449) | 0.743 (0.739, 0.746) | 0.689 (0.684, 0.693) | 0.982 (0.980, 0.983) | 0.604 (0.599, 0.609) |
| >12–18 Hours | 0.990 (0.988, 0.992) | 0.312 (0.308, 0.317) | 0.548 (0.544, 0.552) | 0.433 (0.429, 0.438) | 0.994 (0.993, 0.995) | 0.475 (0.471, 0.480) |
| >18–24 Hours | 0.850 (0.843, 0.857) | 0.552 (0.545, 0.560) | 0.827 (0.823, 0.830) | 0.821 (0.817, 0.824) | 0.955 (0.952, 0.957) | 0.670 (0.664, 0.675) |
| >18–24 Hours | 0.900 (0.894, 0.906) | 0.490 (0.483, 0.497) | 0.786 (0.782, 0.789) | 0.756 (0.752, 0.760) | 0.967 (0.965, 0.969) | 0.635 (0.629, 0.640) |
| >18–24 Hours | 0.950 (0.946, 0.954) | 0.398 (0.392, 0.404) | 0.693 (0.689, 0.697) | 0.626 (0.621, 0.630) | 0.980 (0.978, 0.981) | 0.561 (0.556, 0.566) |
| >18–24 Hours | 0.990 (0.988, 0.992) | 0.280 (0.276, 0.285) | 0.473 (0.469, 0.478) | 0.339 (0.334, 0.343) | 0.992 (0.991, 0.994) | 0.437 (0.433, 0.442) |
| >24–30 Hours | 0.850 (0.843, 0.857) | 0.515 (0.507, 0.522) | 0.803 (0.799, 0.806) | 0.790 (0.786, 0.794) | 0.953 (0.950, 0.955) | 0.641 (0.635, 0.647) |
| >24–30 Hours | 0.900 (0.894, 0.906) | 0.466 (0.459, 0.473) | 0.766 (0.762, 0.769) | 0.730 (0.726, 0.735) | 0.965 (0.963, 0.968) | 0.614 (0.609, 0.620) |
| >24–30 Hours | 0.950 (0.946, 0.954) | 0.383 (0.377, 0.389) | 0.673 (0.668, 0.677) | 0.600 (0.595, 0.605) | 0.979 (0.977, 0.980) | 0.546 (0.541, 0.551) |
| >24–30 Hours | 0.990 (0.988, 0.992) | 0.282 (0.277, 0.287) | 0.475 (0.471, 0.479) | 0.340 (0.336, 0.345) | 0.993 (0.991, 0.994) | 0.439 (0.434, 0.443) |

1. Sensitivity = true positive = cared for in the ICU.

2. Precision = Positive predictive value = true positives/[true positive + false positive]. Number needed to evaluate = 1/precision.

3. Accuracy: (true positives + true negatives)/(positives + negatives).

4. Specificity = true negatives = cared for in routine care.

5. Negative predictive value = true negatives/[true negatives + false negatives]

6. The F1 score is a measure of accuracy with a maximum score of 1. It is the harmonic mean of precision and sensitivity.

The identification of true positives is most relevant to identifying those patients expected to transfer to the ICU from routine care areas or remain in the ICU. The data shown are the estimates and 95% confidence intervals (CI).

analysis of the frequency of importance of model covariates for ICU predictions which found differences in several specific medication classes and coma score between the single-site and the multi-institutional models. These differences were consistent with unique practice patterns at the single site where intubated children cared for in the ICU always receive medication categories of multiple lubricants and irrigations and may receive more anti-psychotic medications due to enhanced delirium screening protocols. Additionally, in the single site, they have the unique practice of requiring all children less than 3 years old requiring continuous bronchodilator therapy to be automatically admitted to the ICU in addition to all patients requiring hourly neurologic checks.

a.

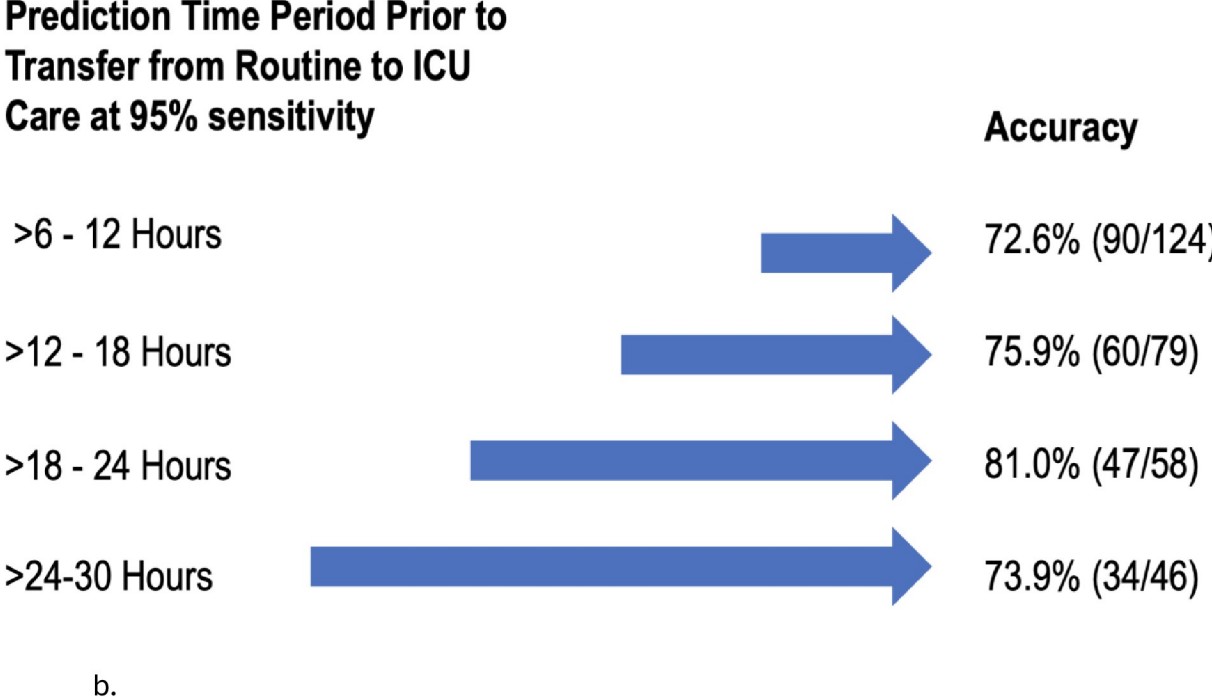

b.

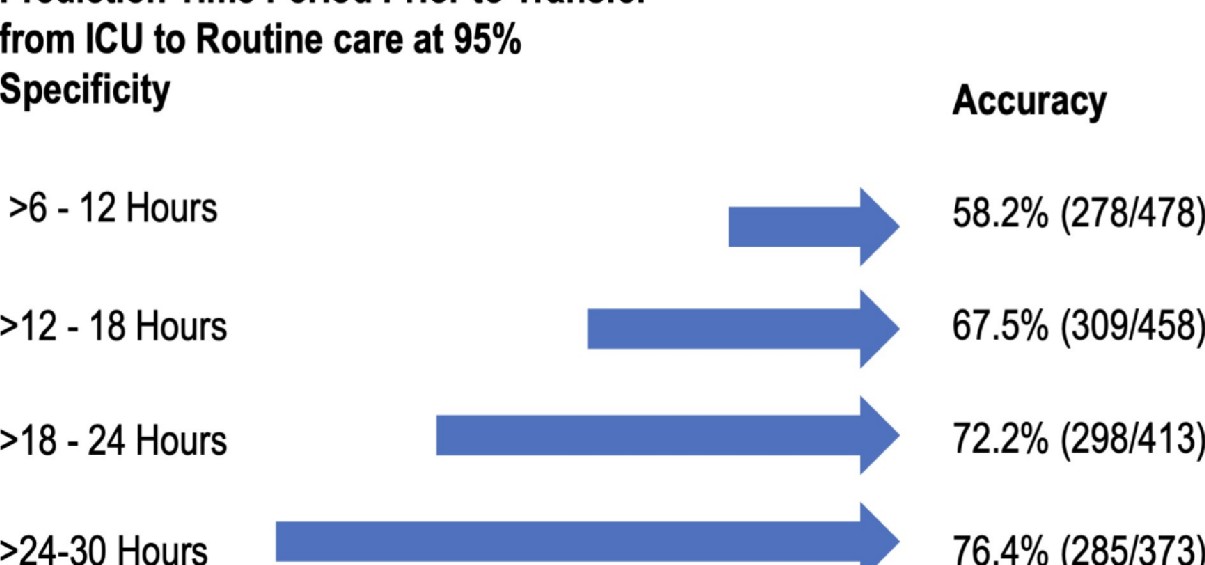

**Fig 4.** Percent accuracy of predictions for patients in the test sample who transferred from routine care to ICU care (n = 124) (a), and from ICU to routine care (n = 478) (b) for the single-site Models of the Criticality Index-Dynamic. The denominator varies by model because patients were required to receive care in a location for a minimum of 6 hours prior to transfer and have ≥1 new data element for a new prediction to be made.

Since mortality is rare in children, pediatric studies may often need other, more common outcomes, which may be influenced by local factors, emphasizing the importance of site-specific testing and re-calibration. However, our experience indicates that the methods developed in national models may be directly applicable to local efforts.

The performance of the single-site CI-D models indicates a potential for clinical trials. For example, at a sensitivity of 0.95, the time period of 6–12 hours prior to transfer had a specificity of 0.561, a negative predictive value of 0.995, and a number needed to evaluate of 2. Even at 24–30 hours prior to transfer at a sensitivity of 0.95, the specificity was 0.340, negative predictive value was 0.978, and number needed to evaluate was 2.6, demonstrating potential clinical applicability with minimal alarm fatigue. While there was a decrement in performance as the prediction time period lengthened, performance remained in the clinically useful range.

This is the first pediatric site-specific assessment of a predictor of change in inpatient clinical status developed from a large multi-institutional dataset using machine learning. With the rapidly increasing availability of very large, multi-institutional datasets, it is likely that similar predictors will evolve with performance characteristics that suggest they are ready for clinical use. Our results suggest that models developed from multi-institutional datasets should be carefully assessed locally prior to clinical use and may benefit from re-development with site-specific data prior to clinical deployment. However, it is likely that the multi-institutional efforts including the variable set and modeling methods will be directly applicable to the re-development of local models. This is a fundamentally different approach than predictors, such as the Pediatric Risk of Mortality score, that compare single-site results to national datasets to evaluate performance relative to multi-institutional benchmarks [22, 23].

The PEWS is currently the primary method of determining patients at risk for needing ICU-level care based on a limited set of five nurse-collected variables [4, 24]. Since its inception, the PEWS quickly disseminated into widespread use despite a notable lack of precision with a number needed to evaluate of >17 in the original model [24]. Importantly, use of the PEWS score was not associated with an improvement in hospital outcome when tested in a multi-center effectiveness study [3].

Recent advances in artificial intelligence using information from health records and other sources offer substantial promise for improvements in clinical assessments [25]. Although potential harm may result from use of inadequately trained or assessed models [26], there are no generally accepted guidelines addressing machine learning models analogous to the Transparent Reporting of Multivariable Prediction Model for Individual Prognosis guidelines [27, 28]. Proposals such as the "Model Facts" label could advance the appropriate clinical use of machine leaning by providing insight into clinically-relevant decision thresholds, precise definition of the outcome, and patient populations that models are designed for [29].

## Limitations

This study has several limitations. First, we did not attempt to optimize performance in the single-site models, including testing additional variables, evaluating different prediction time intervals, and exploring different machine learning methodologies. Second, the dataset used to develop the single-site CI-D models included admissions through the Emergency Department, emphasizing a patient population with acute illness. We selected this sample to maximize the proportion of patients with uncertain status that could worsen. Third, although machine learning methods have the advantage of detecting complex interactions and associations in large datasets, the deep neural network models do not directly provide information on variable importance. The significance of individual or groups of variables is often difficult to ascertain, even with explainable artificial intelligence methods, potentially hindering widespread

acceptance [30]. Advancements in explainable artificial intelligence are anticipated to improve this issue [31, 32].

## Conclusion

Our results suggest that models developed from multi-institutional datasets and intended for application in single-institutions should be assessed locally and may benefit from re-development with site-specific data prior to clinical deployment. The compendium of CI-D models is a promising advancement in early detection of patients who are at risk for needing ICU care, remaining in the ICU, as well as identifying those who could be assessed for transfer from the ICU to routine care.

## Supporting information

**S1 File. Independent variables for Institutional Criticality Index-dynamic models.** All laboratory, vital sign, and demographic variables are outlined in S1 File. A summary of medication data is provided, further elaborated in S2 File.
(PDF)

**S2 File. Medication data for Institutional Criticality Index-dynamic models.** Details of medication classification are included in S2 File.
(PDF)

**S3 File. Details of imputation for Institutional Criticality Index-dynamic models.** All laboratory and vital sign data values imputed in development and validation of the Criticality Index are included in S3 File.
(PDF)

**S4 File. Accuracy of transitions from routine to ICU care at fixed sensitivity thresholds and ICU to routine care at fixed specificity thresholds.** Percent accuracy of predictions for patients in the test sample who transferred from routine care to ICU care (n = 124) at fixed sensitivities of 0.85, 0.90, and 0.99 (a), and from ICU to routine care (n = 478) at fixed specificities of 0.85, 0.90, and 0.99 (b) for the Institutional Models of the Criticality Index-Dynamic.
(PDF)

## Author Contributions

**Conceptualization:** Anita K. Patel, Eduardo Trujillo-Rivera, James M. Chamberlain, Hiroki Morizono, Murray M. Pollack.

**Data curation:** Anita K. Patel, Eduardo Trujillo-Rivera.

**Formal analysis:** Anita K. Patel, Eduardo Trujillo-Rivera, Murray M. Pollack.

**Funding acquisition:** Anita K. Patel.

**Investigation:** Anita K. Patel, Hiroki Morizono, Murray M. Pollack.

**Methodology:** Anita K. Patel, Eduardo Trujillo-Rivera, James M. Chamberlain, Hiroki Morizono, Murray M. Pollack.

**Project administration:** Anita K. Patel, Murray M. Pollack.

**Resources:** Anita K. Patel, Hiroki Morizono, Murray M. Pollack.

**Software:** Anita K. Patel, Eduardo Trujillo-Rivera.

**Supervision:** Anita K. Patel, James M. Chamberlain, Hiroki Morizono, Murray M. Pollack.

**Validation:** Anita K. Patel, Eduardo Trujillo-Rivera, James M. Chamberlain, Murray M. Pollack.

**Visualization:** Anita K. Patel.

**Writing – original draft:** Anita K. Patel, Murray M. Pollack.

**Writing – review & editing:** Anita K. Patel, Eduardo Trujillo-Rivera, James M. Chamberlain, Hiroki Morizono, Murray M. Pollack.

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
