## [Decision Letter · Decision Letter 0]

6 Jul 2022

PONE-D-22-13708External Evaluation of the Dynamic Criticality Index: A Machine Learning Model to Predict Future Need for ICU Care in Hospitalized Pediatric Patients.PLOS ONE

Dear Dr. Patel,

Thank you for submitting your manuscript to PLOS ONE. After careful consideration, we feel that it has merit but does not fully meet PLOS ONE’s publication criteria as it currently stands. Therefore, we invite you to submit a revised version of the manuscript that addresses the points raised during the review process.

Please pay close attention to the methodology clarifications raised by the reviewers.==============================

We look forward to receiving your revised manuscript.

Kind regards,

Bobak J. Mortazavi, PhD

Academic Editor

PLOS ONE

Journal Requirements:

Reviewers' comments:

Reviewer's Responses to Questions

**Comments to the Author**

1. Is the manuscript technically sound, and do the data support the conclusions?

Reviewer #1: Partly

Reviewer #2: Yes

Reviewer #3: Yes

2. Has the statistical analysis been performed appropriately and rigorously? 

Reviewer #1: No

Reviewer #2: Yes

Reviewer #3: No

3. Have the authors made all data underlying the findings in their manuscript fully available?

Reviewer #1: No

Reviewer #2: No

Reviewer #3: Yes

4. Is the manuscript presented in an intelligible fashion and written in standard English?

Reviewer #1: Yes

Reviewer #2: Yes

Reviewer #3: Yes

5. Review Comments to the Author

Reviewer #1: The authors compared the performance of an existing model with a customized model on an independent center's healthcare data, for predicting need for ICU care in hospitalized pediatric patients. While the exercise is interesting and signals caution for applying existing models for local clinical utility, there is somewhat weak support for the implication that the authors wish to draw based on their study. Some comments include:

- When comparing performance of the existing model with the new model, they should be evaluated on the same cohort. So if the new model is evaluated on a test set, the existing model should be evaluated on that set too.

- There is always a need to recalibrate the model, if there is difference in prevalence of the outcome between the development dataset and the test set. Otherwise the performance metrics based on a threshold on the probabilistic output will be off. This is expected.

- The authors concluded that while the model developed on multi-institutional data has poor performance, the methods can be used to develop customized local models. However, a large number of variables were included in the model and the original model does not go through variable selection and it is unclear whether the architecture of the model was optimized for the problem. Therefore, it looks like the single center model can be developed without borrowing any knowledge from the existing model. What then is the value of the multi-institutional model?

- The authors also mentioned that the degradation of performance is probably due to hospital characteristics. Then the original model should include such information in the model and make the model adaptable to different hospital characteristics. Would this improve the performance on a single center data, adjusting for the center's characteristics? It is questionable that the multi-institutional model yields almost no discrimination for the single center's data, whether anything meaningful has been learned or is it completely overfitting? Without digging into these issues and truly understands the root cause for performance degradation, the study is not going to produce much information for future investigations.

Reviewer #2: The authors evaluate whether a machine learning model previously developed from a multi-institutional database will transfer to a new site. The outcome of interest is whether hospitalized children will need ICU care over a variety of time horizons.

The authors find that directly applying the model to the new site leads to very poor performance across a variety of metrics. They do find, however, that the model development process transfers quite well. By following the same development procedure (e.g. variable selection, model design, etc.) of the multi-institutional model, the authors are able to develop a model that performs well when trained with data from the new site. This demonstrates that even when the model itself may not transfer across different sites, the modeling methodology may transfer effectively.

The paper is well-written and the motivation of the work is clear. The presentation of results is comprehensive and methodologically sound.

The problem of robustness across different institutions is a key challenge in the development and deployment of ML models for healthcare applications. The finding that the modeling methodology may transfer effectively even if the model itself does not is an interesting one that has practical relevance.

I realize that a significant focus of this work was to replicate the methodology of prior work which determined the modeling decisions. However, I think it would still be beneficial to see a comparison with standard baselines such as logistic regression and decision tree algorithms (e.g. XGBoost) to put the results into context. Is the neural network necessary for strong performance? The authors acknowledge the drawbacks of utilizing a neural network (e.g. interpretability), but do not demonstrate whether there are benefits to using deep learning instead of other approaches.

Given that a neural network is being used, it would also be valuable to report full implementation details to aid in replication, even if the information is relegated to the appendix. The only details I see are that it is a fully connected network with 5 hidden layers. How many hidden units were used? How was the network trained (optimizer, learning rate, etc.)? Were regularization techniques utilized (l2 regularization, dropout, etc.)? How were the hyperparameters selected?

There is some discussion of why the model may have failed to transfer, but it would be valuable to gain more insight into this. For example, the calibration plot shows that the multi-institutional model tends to be underconfident in the new setting. Is there any insight into why this may be the case?

I may be missing something, but in Figure 1a (> 12 − 18 Hours) the AUROC is reported as .493, but the ROC curve appears to lie above the dashed y=x line. Shouldn't the AUROC be >0.5 in that case?

Reviewer #3: Strength:

• This study is based on a decent dataset with a great number of examples. The authors provide detailed information about the dataset (Table 1), including admissions, age, gender, etc.

• The authors provide great metrics to evaluate their experiment, including sensitivity, precision, accuracy, specificity, negative predictive value, F1, as well as AUROC and AUPRC, giving a comprehensive understanding of the model performance.

Weakness:

• Experiment setting lack details. The authors mention that 75% patients are randomly selected as training data, and 13% for validation and 12% for test (page 6). Please declare how many repeated experiments (or cross validation? I assume not) have been done.

• Need more explanation of data and data preprocessing. The authors also declare that “missing data were imputed” (page 7). Please add more details of how are the missing data imputed (zeros, previous values, average values, etc.). In addition, what is the frequency of the data? What features are chosen? Please make them clear in the paper.

• The paper lacks background information and related work. Please add some citations about others methods solving this problem, and please explain why it is important to evaluate the CI-D model on individual sites.

• The paper needs some baseline model to compare with. I understand that this paper is an evaluation of a previously proposed model. However, only testing CI-D model is not convincing. The authors should introduce some other method to compare with CI-D model to prove that CI-D is or is not working.

• The result in Table 2.A seems poor, e.g., F1 scores are below 0.02. The authors also mention this (the first paragraph of page 10). Please have some explanation or analysis of why the performance is very low here.

• In figure 1b, how is it possible that the precision-recall curve decrease at the beginning and then increase (the valley of precision values below 0.6 when recall is around 0). Please check the experiment results and have an explanation of that.

• It is not very easy to understand the table 2 and 3. Please have some highlights of the results.

6. PLOS authors have the option to publish the peer review history of their article (what does this mean?). If published, this will include your full peer review and any attached files.

Reviewer #1: No

Reviewer #2: No

Reviewer #3: **Yes: **Lida Zhang

---

## [Author Response · Author response to Decision Letter 0]

6 Oct 2022

Dear Editor-in-Chief,

I am re-submitting the original research article titled: “External Evaluation of the Dynamic Criticality Index: A Machine Learning Model to Predict Future Need for ICU Care in Hospitalized Pediatric Patients” by Patel et. al after an extensive revision. We appreciate your consideration for publication in PLOS ONE and the thoughtful editorial review that we believe has now greatly strengthened this manuscript. This will be the first ever publication to externally evaluate a pediatric machine learning algorithm developed from multi-institutional data that predicts future inpatient care needs in a single site; and compare these results to a single-site model developed using identical modelling methods and variables.

We believe our research is best suited for the PLOS ONE because the knowledge garnered from this research can positively influence and inform sound methods for evaluation of pediatric prediction algorithms that use novel machine learning methodology. This can have particularly important implications for models developed and intended for future clinical use as our results suggest that models developed from multi-institutional datasets should be assessed at individual institutions and may benefit from re-development with site-specific data prior to clinical deployment

We confirm that this manuscript has not been published elsewhere, is not under consideration by another journal, nor has the data been presented in another format such as abstract/poster/presentation. All authors have participated in the concept and design, analysis, and interpretation of data, in addition to drafting, reviewing, and/or revision of the manuscript. All authors have approved the final manuscript and agree with its submission to your journal.

A point by point response to reviewers is included below.

Please let me know of your decision at your earliest convenience.

With my best regards,

Anita K Patel, MD

Corresponding Author

Department of Pediatrics, Division of Pediatric Critical Care, Children’s National Health System and the George Washington University School of Medicine and Health Sciences, Washington DC

111 Michigan Ave NW

Washington, DC 20011

apatel4@childrensnational.org

202-476-6817

Reviewer #1: The authors compared the performance of an existing model with a customized model on an independent center's healthcare data, for predicting need for ICU care in hospitalized pediatric patients. While the exercise is interesting and signals caution for applying existing models for local clinical utility, there is somewhat weak support for the implication that the authors wish to draw based on their study. Some comments include:

- When comparing performance of the existing model with the new model, they should be evaluated on the same cohort. So if the new model is evaluated on a test set, the existing model should be evaluated on that set too.

This was an excellent suggestion; therefore, we revised the performance comparison by applying the original multi-institutional model to only the test set of the single-institution dataset. Now the comparative assessments compare the multi-institutional models’ performance with the institution specific models’ performances when applied to the test set of the institution specific dataset.

Initially, we chose to apply the multi-institutional model to the full single-site dataset instead of using the test set only because the test set is a random subsample of the full dataset. As the multi-institutional model was developed on a different data, there is no risk of overfitting, and the performance should be similar in either the full dataset or a random subsample. However, for model comparison we believe the reviewer’s suggestion would improve the paper’s overall results and conclusion. Therefore, Table 2, Figures 1a-c and result’s paragraph two have now been updated with the analysis of the multi-institutional model applied to the single-site test dataset. The multi-institutional model’s performance on test single-site dataset and full single-site dataset were similar, but confidence intervals are wider on the test dataset.

- There is always a need to recalibrate the model, if there is difference in prevalence of the outcome dataset and the test between the development set. Otherwise the performance metrics based on a threshold on the probabilistic output will be off. This is expected.

Thank you. There is no substantial difference in prevalence between the multi-institutional and the single-site dataset. The original multi-institutional model performance detailed in this study (lines 169-175) were from the simulated children’s hospital reported in the original paper with an ICU prevalence rate of 20%, similar to the prevalence rate in this single-site sample which was 22.5%. Therefore, we believe the comparison, and conclusion of loss of performance was accurate. In this case, the need for recalibration is not because of differences in prevalence. For clarity, we have now emphasized the congruent prevalence rates in line 169: “Performance of the multi-institutional CI-D applied to the single-site test dataset was compared to the multi-institutional CI-D models’ performances applied to a previously reported simulated children’s hospital,(10) consisting of all 7,054 pediatric admissions in the Health Facts� database from January 2016 to June 2018 with an ICU prevalence rate of 20.0%, similar to the single site sample of 22.5%.(10)”

- The authors concluded that while the model developed on multi-institutional data has poor performance, the methods can be used to develop customized local models. However, a large number of variables were included in the model and the original model does not go through variable selection and it is unclear whether the architecture of the model was optimized for the problem. Therefore, it looks like the single center model can be developed without borrowing any knowledge from the existing model. What then is the value of the multi-institutional model?

The original multi-institutional models enabled us to develop generalizable models that could be re-developed using site-specific data in a separate institution. The power of developing a national model first is that the variables and modeling methodology are generalizable to most hospitals that care for children. If we developed a site-specific model first, the variable selection would be skewed to only detect patients at the single institution. This research also had a primary aim of evaluating the performance of a national model predicting ICU or inpatient care needs at an individual site and if it could be directly applied to a single institution. Because our results suggested that model performance had degraded, we came to a generalizable conclusion that the national algorithm, using identical variables and model architecture, could be locally adapted at a single institution for potential clinical use. The paper’s conclusion suggests that national model’s utility is in developing a generalizable framework that can be used at a wide variety of hospitals. This conclusion is not trivial because the data science expertise is variable in different institutions and there may be care nuances requiring institution specific variables. We have also emphasized the identical variable selection and model architecture in the methods – machine learning methodology section of the paper (lines 178-180):

“The machine learning methodology for the single-site CI-D models from the single-site data set followed the identical process as that of the multi-institutional models with identical variables and congruent model architecture.(8),(10)” 

The authors also mentioned that the degradation of performance is probably due to hospital characteristics. Then the original model should include such information in the model and make the model adaptable to different hospital characteristics. Would this improve the performance on a single center data, adjusting for the center's characteristics? It is questionable that the multi-institutional model yields almost no discrimination for the single center's data, whether anything meaningful has been learned or is it completely overfitting? Without digging into these issues and truly understands the root cause for performance degradation, the study is not going to produce much information for future investigations.

The hospital characteristics we were referring to were primarily related to local nursing and resource utilization practices that differ among hospitals and cannot be captured in national datasets. For example, some hospitals have smaller nursing to patient ratios in inpatient units that allow for more resource intensive patients to be cared for in these units. Whereas, other institutions must admit patients to the ICU simply for enhanced nursing care needs. Such intangible attributes are frequently not available in national databases. With regard to overfitting, we provided metrics in the original paper (REF #18) to support that the model was rigorously evaluated and was not overfitted. We purposefully did not include hospital characteristics (such as bed size and region) in the multi-institutional model to focus on capturing intensity of treatment and physiological status for predicting the outcome. Our current supposition for the degradation in performance in the multi-institutional model applied to the single-site dataset is different relationships between predictors and the outcome of ICU admission that would be reflected in variable importance, To address this question we performed a variable importance analysis using the Local Interpretable Model-Agnostic Explanation (LIME) approach (LIME R package).(Pedersen and Benesty, 2021) to determine if the differing variables in the congruent models could explain practice differences reflecting differing institutional policies/practices. The methodology for this analysis has been added to methods lines 234-258:

“We explored the variable importance using a Local Interpretable Model-Agnostic Explanation (LIME) approach (LIME R package) to support the possibility that practice differences including ICU admission criteria could, in part, account for the performance of the multi-institutional models in the single-site.(19) Briefly, the LIME approach is based on the assumption that every model performs like a linear prediction model for each prediction and the hierarchy of covariate importance is preserved. The collection of individual linear models provides an interpretation of the covariate importance in the final model. We ranked the most important covariates across all ICU risk predictions by computing the percentage of times each variable was among the 30 most important covariates. We evaluated the four single-site machine learning models on the test single-site dataset and the four multi-institutional models on the multi-institutional dataset. We selected the subset of cases where the predicted risk of need for ICU care was between 0.245 and 0.255 (referred to as “risk condition”). We selected a standard risk condition for the purpose of variable importance comparison between models among patients with a similar risk of ICU admission. Using this risk condition, in the single-site test dataset, a total of 201-202 cases were assessed per prediction model, and between 953 and 2,328 cases were assessed from the multi-institutional test set. We focused on covariates that might indicate ICU practices differences and used the following as indicators of practice differences: medications and coma scores. Medications were focused on because care practices at some institutions favor the use of specific medications in their ICU patients that are rarely administered on the inpatient unit. Coma score was identified because the ability to care for children with altered states of consciousness in non-ICU areas varies among institutions.” 

The following paragraph has been added to the results section lines 632-651: 

“An assessment of covariate importance comparing the single-site models and the multi-institutional models using the LIME approach is presented in Figure 3. Among the full list of covariates, 20%-30% of the 30 most frequent covariates used for ICU risk predictions made in the multi-institutional models were not among the 30 most frequent variables in the single-site models. Similarly, between 17%-25% of the most frequent covariates used for ICU predictions in the single-site models were not among the 30 most frequent covariates used in the multi-institutional models. Certain medications and coma score were more frequently in the 30 most frequent single-site covariates, consistent with unique practices patterns in this site. For example, the proportion of time on antipsychotics, bronchodilators, and number of lubricants/irrigations were among the 30 most frequently utilized covariates in the single-site model, but not in the multi-institutional model. Multiple nasal lubricants and anti-psychotics are medication categories commonly administered to invasively ventilated patients at the single-site and all children less than 3 years old requiring continuous bronchodilator therapy must be automatically admitted to the ICU, practices that may not be shared at other institutions but are unique to the single-site. Average coma score was among the 30 most frequently used covariates in the single-site model, but not in the multi-institutional model. Similarly, minimum and maximum coma score were more frequently among the 30 most frequent covariates used in the single-site model when compared to the multi-institutional model (3.4% in the single-site model vs 2.5% in the multi-institutional model, and 3.4% in single-site model vs 1.2% in multi-institutional model respectively). The single-site has a policy which requires all patients on every one-hour neurologic assessments to be monitored in the ICU; this practice is not shared at all institutions and may account for coma score values cared for in the single-site ICU to be different than in other institutions.”

The following lines have been added to discussion paragraph one, lines 717-725:

“This possibility was supported by the LIME analysis of the frequency of importance of model covariates for ICU predictions which found differences in several specific medication classes and coma score between the single-site and the multi-institutional models. These differences were consistent with unique practice patterns at the single site where intubated children cared for in the ICU always receive medication categories of multiple lubricants and irrigations and may receive more anti-psychotic medications due to enhanced delirium screening protocols. Additionally, in the single site, they have the unique practice of requiring all children less than 3 years old requiring continuous bronchodilator therapy to be automatically admitted to the ICU in addition to all patients requiring hourly neurologic checks.” 

Reviewer #2: The authors evaluate whether a machine learning model previously developed from a multi-institutional database will transfer to a new site. The outcome of interest is whether hospitalized children will need ICU care over a variety of time horizons.

The authors find that directly applying the model to the new site leads to very poor performance across a variety of metrics. They do find, however, that the model development process transfers quite well. By following the same development procedure (e.g. variable selection, model design, etc.) of the multi-institutional model, the authors are able to develop a model that performs well when trained with data from the new site. This demonstrates that even when the model itself may not transfer across different sites, the modeling methodology may transfer effectively.

The paper is well-written and the motivation of the work is clear. The presentation of results is comprehensive and methodologically sound.

The problem of robustness across different institutions is a key challenge in the development and deployment of ML models for healthcare applications. The finding that the modeling methodology may transfer effectively even if the model itself does not is an interesting one that has practical relevance.

I realize that a significant focus of this work was to replicate the methodology of prior work which determined the modeling decisions. However, I think it would still be beneficial to see a comparison with standard baselines such as logistic regression and decision tree algorithms (e.g. XGBoost) to put the results into context. Is the neural network necessary for strong performance? The authors acknowledge the drawbacks of utilizing a neural network (e.g. interpretability), but do not demonstrate whether there are benefits to using deep learning instead of other approaches.

We opted to focus on the primary aims of this research report, which were to 1) apply the original model to an independent single-institution dataset 2) determine if using the same methodology, but retraining the model with single institution data would improve performance. Your suggestion of exploring different types of machine learning models is excellent. Of course, we have tried other modeling approaches at arriving at the current approach, but not in manner that is sufficiently rigorous to publish. We are contemplating a comparison of different approaches for a future study. Furthermore, we believe this would be out of the intended scope of this study to pursue different methods for this manuscript.

Given that a neural network is being used, it would also be valuable to report full implementation details to aid in replication, even if the information is relegated to the appendix. The only details I see are that it is a fully connected network with 5 hidden layers. How many hidden units were used? How was the network trained (optimizer, learning rate, etc.)? Were regularization techniques utilized (l2 regularization, dropout, etc.)? How were the hyperparameters selected?

Thank you for this suggestion, we have added this to the methods section of the paper lines 231-250 as follows:

“A single neural network for binary classification for ICU outcome was developed for each future prediction time-period. The objective of each training epoch was to maximize the Mathew’s correlation coefficient at a cut point of 0.5 while minimizing the binary cross entropy between the predicted score and the patient’s outcome. We started with a neural network with a single hidden layer, and logit output and consecutively increased the number of nodes while monitoring the Mathew correlation coefficient, sensitivity, specificity, precision, negative predictive value at cut points of 0.15, 0.5, and 0.9 in the training and validation sets. If overfitting was found, we add L2 regularization, and layer node dropouts with parameters tuned to maintain similar metrices on the validation and training sets. We increased the number of nodes if there was not overfitting or if it was eliminated using the outlined procedure. When there were no additional gains on the performance metrics , we added another hidden layer and repeated the process, increasing the number of hidden layers until our regularization attempts were unsuccessful in avoiding overfitting. We keep the best model as measured by the Mathew correlation coefficient without overfitting. Glorot uniform was used to initialize each layer’s weight, and optimized using RMSprop. An initial learning rate of 0.0002 was used with minibatches of 10,000.(19) The final neural network was fully connected with five sequential hidden layers, an output layer with one node and logistic activation. Each neural network was calibrated to the future risk of ICU care using B-splines polynomials and logistic regression.”

There is some discussion of why the model may have failed to transfer, but it would be valuable to gain more insight into this. For example, the calibration plot shows that the multi-institutional model tends to be underconfident in the new setting. Is there any insight into why this may be the case?

Thank you for this question, as reviewer one posed a similar one. To address the performance of the multi-institutional model applied to the single-site (test) dataset, we have now added a covariate importance analysis to the paper as outlined above in the final question response to reviewer one. We believe that this suggestion has substantially improved the quality and impact of this paper.

I may be missing something, but in Figure 1a (> 12 − 18 Hours) the AUROC is reported as .493, but the ROC curve appears to lie above the dashed y=x line. Shouldn't the AUROC be >0.5 in that case?

Thank you for pointing out this detail. The numeric value 0.493 is correct, the construction of the curve needed more resolution. The AUC curve is constructed by computing the pairs (sensitivity, specificity) as functions of a threshold >= 0 and <= 1. We computed these pairs in a lattice of threshold values ranging from 0 to 1. The plot is created by joining a line with these pairs. We did not create a sufficiently dense lattice of thresholds to create the AUC curve with sufficient resolution. To address this, we have increased the resolution of this lattice and plotted these pairs again. It can now be seen that the curve does not always lie above the dashed y=x line.

Reviewer #3: Strength:

• This study is based on a decent dataset with a great number of examples. The authors provide detailed information about the dataset (Table 1), including admissions, age, gender, etc.

• The authors provide great metrics to evaluate their experiment, including sensitivity, precision, accuracy, specificity, negative predictive value, F1, as well as AUROC and AUPRC, giving a comprehensive understanding of the model performance.

Weakness:

• Experiment setting lack details. The authors mention that 75% patients are randomly selected as training data, and 13% for validation and 12% for test (page 6). Please declare how many repeated experiments (or cross validation? I assume not) have been done.

There are no cross validations. There were not repeated experiments. To the best of our knowledge, cross validation in this setting is useful to choose the hyperparameters of the model and to avoid overfitting. We did not choose hyperparameters in this way. We have expanded the description of model building to provide a detailed account of our machine learning methodology and process. Please see the response to the reviewer above for a full explanation and lines 231-250 of the methods section.

• Need more explanation of data and data preprocessing. The authors also declare that “missing data were imputed” (page 7). Please add more details of how are the missing data imputed (zeros, previous values, average values, etc.). In addition, what is the frequency of the data? What features are chosen? Please make them clear in the paper.

Details of the imputation are included both in references noted in lines 229-230 by referring to citations 8, 18. and in Appendix 3 (formerly Appendix 1) where we report the percentage frequencies of first time slices where data was required to be imputed with age-related medians. Directly taken from the first paragraph of appendix 3: “ The initial time period required values for all laboratory and vital sign data. We imputed the values from the medians by age groups of those patients who had these measurements in the first time period. Note, that the imputed data also included the measurement count for the laboratory test or vital sign measurement which was set to 0 to indicate an imputed value. The age groups were a composite of the age groups used for display of normal data by various sources.” Additionally, we have now added a new Appendix 1 and 2 detailing all independent variables included in the model, including details of medication classification. 

• The paper lacks background information and related work. Please add some citations about others methods solving this problem, and please explain why it is important to evaluate the CI-D model on individual sites.

Currently there is no similar validated model to the CI-D for pediatrics using machine learning/electronic medical record related data. We do mention in the introduction the only model currently available to assess risk for clinical deterioration, which is the PEWS score (lines 90-92). Therefore, the lack of background on related work is reflective of the current state of early warning scores in pediatrics. It is important to evaluate the CI-D in individual sites to determine if the model can be directly applied to a single institution while maintaining the high performance metrics originally reported. The ultimate goal of the CI-D is to be deployed in hospitals to aid in detecting patients at risk for needing the ICU. Therefore, applying the model to a single institution was the necessary next step prior to potential clinical testing and deployment.

• The paper needs some baseline model to compare with. I understand that this paper is an evaluation of a previously proposed model. However, only testing CI-D model is not convincing. The authors should introduce some other method to compare with CI-D model to prove that CI-D is or is not working.

This is an excellent suggestion – however the only possible comparison to the CI-D is the PEWS score, which includes subjective and objective variables, one of which includes “behavior” that cannot be ascertained from an EHR-related database. Therefore, it was not possible to compare our purely objective, data-driven model to the PEWS score. However, in the next phase of analyses once we have a silent deployment of the model, we plan to compare performance of CI-D to the PEWS score. We did dedicate a paragraph in the discussion (lines 799-804) to detailing the performance of the PEWS, with its very low precision, as a method of comparing the CI-D performance to the PEWS score.

• The result in Table 2.A seems poor, e.g., F1 scores are below 0.02. The authors also mention this (the first paragraph of page 10). Please have some explanation or analysis of why the performance is very low here.

Table 2a presents the derivations of the confusion matrix of the multi-institutional models evaluated on the test CN data set as classifier by using a threshold of 0.5. Table 2b presents the evaluation of the single institution models as a classifier on the single-site test set by using a threshold of 0.5. We chose to use a threshold of 0.5 for classification of both sets of models as a reference standard for model comparison. An alternative to using a fixed threshold would be to compare the models at a fixed sensitivity (similar to the analysis in Table 3 for the single site models). However, if the models are presented in this way, it does not sufficiently reveal the poor ICU predictions in the multi-institutional models which tend to predict that most patients are not admitted to the ICU.

The results in table 2A show poor performance because of degradation when the multi-institutional models are applied to the single-site test set. We contend that the relationship between the predictors and the outcome in multi-institutional and single-site datasets may be different. The multi-institutional model tends to under-predict the risk of ICU admission in the single-site, and we have now provided a covariate feature importance analysis to provide an explanation for this phenomenon. This is extensively detailed in the last response to reviewer one.

• In figure 1b, how is it possible that the precision-recall curve decreases at the beginning and then increase (the valley of precision values below 0.6 when recall is around 0). Please check the experiment results and have an explanation of that.

Thank you. We have changed the curve. It is now evaluated on the single-site test set as requested Reviewer 1. It was previously computed using the entire single-site dataset. The new curve has similar characteristics to those noted by the reviewer.

We believe the Reviewer’s comments are a result of the following: The curve is a collection of observed point estimates from the test sample plotted using the highest resolution grid without redundancy. Given these conditions, the precision fluctuates. However, while we are able to plot the point estimates, we are not able to provide confidence intervals for these data. Confidence intervals, if they could be computed, would explain the fluctuation seen in the plots. 

Hopefully, the reviewer can see with the following examples that the point estimate of the precision fluctuates, resulting in a fluctuating PR curve due to a high resolution threshold, available sample, and risk predictions. For a complete understanding, confidence intervals need to be studied in combination with the point estimates of the associated recalls. 

Case threshold Recall/sensitivity precision

1 Infinity 0 1

2 0.873678 8.84E-05 1

3 0.86991 8.84E-05 0.5

4 0.863507 1.77E-04 0.666667

5 0.860584 2.65E-04 0.75

6 0.857497 3.53E-04 0.8

7 0.854567 4.42E-04 0.833333

8 0.851362 5.30E-04 0.857143

9 0.848431 6.18E-04 0.875

10 0.841927 7.07E-04 0.888889

11 0.829499 7.07E-04 0.8

 … … …

The confusion matrix for case 2 is below:

Reference

Prediction ICU Inpatient

 ICU 1 0

 Inpatient 11317 43486

The described classifier in the table above only predicts one instance to be admitted to ICU. In this case, the precision is 1 / (1+0) = 1. This is the point estimate, but a 95% confidence interval for this ratio should be wide because the denominator is only 1. 

The confusion matrix for case 3 is below:

Reference

Prediction ICU Inpatient

 ICU 1 1

 Inpatient 11317 43485

The classifier in case 3 only predict two instances to be admitted to ICU: one prediction is correct and one prediction is incorrect. The precision of this classifier is 1 / (1+1) = 0.5. A 95% confidence interval for this ratio is again wide.

The classifier in case 4 predicts three instances to be in the ICU, two correct predictions, and one incorrect prediction: precision = 2 / (2+1) = 0.667. The classifier in case 5 predicts four instances to be in the ICU, three correct and one incorrect: precision = 3 / (3+1) = 0.75

This pattern continues for all subsequent case examples.

• It is not very easy to understand the table 2 and 3. Please have some highlights of the results.

We realize that these are data intensive tables. In large part it is a consequence of predicting multiple times in the future. However, we have used this format for presenting data previously (Trujillo Rivera EA, et al. Predicting Future Care Requirements Using Machine Learning for Pediatric Intensive and Routine Care Inpatients. Crit Care Explor. 2021) without reviewer or editorial objection and neither of the other 2 reviewers objected. We believe the results section does highlight the results. In paragraphs 2 and 3 we review the performance metrics of the multi-institutional and single-site models when applied to the singlesite dataset for the purpose of comparison of the models’ performances. First sentence, paragraph 2 states: “The performance metrics of the multi-institutional CI-D models applied to the single-site test dataset were predominantly poor”, and the remainder of the paragraph proceeds to present details of Table 2a. First sentence, paragraph 3 states: “The performance metrics of the single-site CI-D models applied to the single-site test sample (Figure 2, Table 2b) were substantially better than the performance of the multi-institutional model” and similarly proceeds to present the performance metrics of the single-site model applied to the single-site test dataset. In paragraph 5 we highlight the results of Table 3: “Table 3 shows the performance metrics of the single-site model applied to test patients for sensitivities of 0.85, 0.90, 0.95, and 0.99 to display alternative clinical decision thresholds.” As stated, the primary purpose of Table 3 and paragraph 5 are to present different clinical decision thresholds that may be relevant for different clinical purposes or use cases, and review how the other performance metrics change with increasing or decreasing sensitivity. 

We are very willing to present the data in a different way if the reviewer or editorial staff have another suggestion.

---

## [Decision Letter · Decision Letter 1]

13 Dec 2022

PONE-D-22-13708R1External Evaluation of the Dynamic Criticality Index: A Machine Learning Model to Predict Future Need for ICU Care in Hospitalized Pediatric Patients.PLOS ONE

Dear Dr. Patel,

Thank you for submitting your manuscript to PLOS ONE. After careful consideration, we feel that it has merit but does not fully meet PLOS ONE’s publication criteria as it currently stands. Therefore, we invite you to submit a revised version of the manuscript that addresses the points raised during the review process.

 Please address reviewer comments on the revision approach.

We look forward to receiving your revised manuscript.

Kind regards,

Bobak J. Mortazavi, PhD

Academic Editor

PLOS ONE

Reviewers' comments:

Reviewer's Responses to Questions

**Comments to the Author**

1. If the authors have adequately addressed your comments raised in a previous round of review and you feel that this manuscript is now acceptable for publication, you may indicate that here to bypass the “Comments to the Author” section, enter your conflict of interest statement in the “Confidential to Editor” section, and submit your "Accept" recommendation.

Reviewer #1: All comments have been addressed

Reviewer #3: (No Response)

2. Is the manuscript technically sound, and do the data support the conclusions?

Reviewer #1: (No Response)

Reviewer #3: Partly

3. Has the statistical analysis been performed appropriately and rigorously? 

Reviewer #1: (No Response)

Reviewer #3: (No Response)

4. Have the authors made all data underlying the findings in their manuscript fully available?

Reviewer #1: (No Response)

Reviewer #3: Yes

5. Is the manuscript presented in an intelligible fashion and written in standard English?

Reviewer #1: (No Response)

Reviewer #3: Yes

6. Review Comments to the Author

Reviewer #1: (No Response)

Reviewer #3: 1. The goal of the paper is evaluating CI-D model, and assessing the single site performance of CI-D developed from a multi-institutional data base, and introduce neural networks to predict future need for ICU care. Is the neural network intended to help evaluate CI-D model?

2. The contribution of the paper is confusing. It’s okay to evaluate CI-D model and tell readers where does CI-D model perform well and where does not, however, what can reader learn from the neural network? Is it a new proposed method to evaluate another algorithm? Please modify the introduction and declare the contributions.

3. The machine learning model and the application part has flaws and are not well explained:

a) Why there are four separate model, and why these four intervals? (I believe there is some medical support behind, but please emphasize in the paper).

b) What are the neural networks predicting? Is the patient outcome a classification problem or regression (I assume classification because AUROC and AUPRC are the evaluation metrics, but please make it clear in the paper).

c) The authors declared that the objective of teach training epoch was to maximize the Mathew correlation coefficient and minimize the cross entropy between predicted score and the patient’s outcome, however, it’s still not clear what is the loss/cost function? Is it cross entropy only, or a joint loss of cross entropy and (reversed) correlation coefficient?

d) What does the data look like. Is it time-series? Discrete numeric features? other type of features?

e) What exactly is the structure of the neural network? The authors started with a neural network with a single hidden layer, and tried many things later, however, did not declare their decision of the network, nor did they show which attempt helped and which did not.

f) What neural network model is applied? Fully-connected network? RNN/LSTM? CNN?

4. We previous suggested the authors to add other models to compare with: a). Traditional machine learning such as random forest, logistic regression. b). LSTM if time-series data. This is because neural networks do not always perform better than traditional machine learning models. Also, if the data is time-series, LSTM is a better neural network than fully-connected networks.

5. The way to design the network is ad-hoc. The authors tried a few random things and search which works the best, however, is not an ideal way of designing neural networks. The design of a network needs reasons and domain knowledge support. Trying many things cannot tell readers any useful information when they want to apply the paper to their own work, and there are always new techniques to try.

7. PLOS authors have the option to publish the peer review history of their article (what does this mean?). If published, this will include your full peer review and any attached files.

Reviewer #1: No

Reviewer #3: **Yes: **Lida Zhang

---

## [Author Response · Author response to Decision Letter 1]

14 Apr 2023

We are pleased that we seemed to have successfully responded to Reviewers 1 and 2 and hope that these comments are satisfactory to Reviewer 3.

Please note that the overall goal of this analysis is to demonstrate that multi-institutional databases, often considered the ‘gold standard” for models, may loose performance in single sites. This performance issue can be overcome by applying the modeling lessons learned from multi-institutional models but applied to single sites. Reviewer #3 is focusing on more basic modeling issues which have been explored in manuscript references 8, 9, 10, and 18, not the issues related to bias secondary to the databases.

Reviewer #3: 

1. The goal of the paper is evaluating CI-D model, and assessing the single site performance of CI-D developed from a multi-institutional data base, and introduce neural networks to predict future need for ICU care. Is the neural network intended to help evaluate CI-D model?

We are sorry for this reviewer’s confusion. The goal of the manuscript is stated in the last sentence of the introduction (with minor edits, see below): “We performed this study to 1) assess the performance of the CI-D models developed from a multi-institutional database to predict future care locations when applied to a single site that did not participate in the multi-institutional dataset, (10) and 2) to apply the identical variables and modeling methods to a single-site dataset to develop models optimized for the single site, and compare this performance to the multi-institutional CI-D models’ performances.’

The purpose of the manuscript is not to "introduce neural networks to predict future need for ICU care.” Neural networks are the methodology used to derive all models used in the manuscript. To clarify the reviewer’s question, “Is the neural network intended to help evaluate CI-D model?” we did the following: 

1. There is confusion between the use of the terms models, neural networks and algorithms. We eliminated the term “algorithms” and replaced it with “models” throughout the manuscript as it is unnecessarily confusing. 

2. We clarified the issue of model and neural network as follows: Each prediction time-horizon (6 hrs – 12 hrs, etc.) has its own model, developed from separate neural networks. This is noted in Introduction>Para 2> “Four separate models were developed for prediction of care location in each of the four future time intervals.” To emphasize and clarify this issue that is confusing, we (a) revised Intro>Para 2>sentence 1 to: “Recently, we developed a set of four models developed from separate neural networks that predict future care needs assessed as care location using a new severity measure, the Dynamic Criticality Index (CI-D).(8),(9),(10)” (b) We added “models were” to Intro>Para 2> Sentence 3. (c) The last sentence of this paragraph has been clarified as “four models developed from separate neural networks.” (d). The last sentence of the Introduction now reads: “ We performed this study to 1) assess the performance of the CI-D models developed from a multi-institutional database to predict future care locations when applied to a single site that did not participate in the multi-institutional dataset,(10) and 2) to apply the identical variables and modeling methods to a single-site dataset to develop models optimized for the single site, and compare this performance to the multi-institutional CI-D models’ performances.” 

2. The contribution of the paper is confusing. It’s okay to evaluate CI-D model and tell readers where does CI-D model perform well and where does not, however, what can reader learn from the neural network? Is it a new proposed method to evaluate another algorithm? Please modify the introduction and declare the contributions.

Please see response to #1 and General Comments. We hope the revisions proposed in #1 are helpful. Additionally, we have amended the final sentence of the introduction for clarity: “We performed this study to 1) assess the performance of the CI-D models developed from a multi-institutional database to predict future care locations when applied to a single site that did not participate in the multi-institutional dataset,(10) and 2) to apply the identical variables and modeling methods to a single-site dataset to develop models optimized for the single site, and compare this performance to the multi-institutional CI-D models’ performances.”

3. The machine learning model and the application part has flaws and are not well explained:

a) Why there are four separate model, and why these four intervals? (I believe there is some medical support behind, but please emphasize in the paper).

There are four separate models developed using neural networks to predict care needs represented by care locations (ICU and non-ICU) in 4 different time periods in the future (0-6, 6-12, 12-18. 18-24 hours in the future). There are four models for each future prediction because that method had optimal prediction performance. We have now added the following statements for clarity:

a) To address the four 6-hour time period choice, paragraph one of machine learning methodology includes: “Six hours was selected because data acquisition for non-ICU care patients is relatively infrequent compared with ICU patients. A 6-hour time periods allowed for sufficient time to obtain a set of vital signs, and frequently a medication and/or lab result to inform future care location predictions.”

b) To address the relevance of the 4 future time periods and four separate models, the following has been added to paragraph 2 of the machine learning methodology section: “The four future time periods were chosen as clinically relevant time periods for both immediate and future resource allocation planning for potential increases and decreases in patient care needs. Separate models for each time period were used to optimize performance.”

c) To address the issue of the four separate models, we added the phrase “to optimize performance for each time period” to the first sentence of Machine Learning>Para 2 which now reads: “Independent neural networks calibrated to risk of ICU care were developed for four future time periods (>6-12 hours, >12-18 hours, >18-24 hours, and >24-30 hours) to optimize performance for each time period.”

b) What are the neural networks predicting? Is the patient outcome a classification problem or regression (I assume classification because AUROC and AUPRC are the evaluation metrics, but please make it clear in the paper).

We believe that readers with medical knowledge will not have this issue. However, we further clarified this issue. The second paragraph of the section ‘Machine learning methodology’ includes the following sentence clarifying that that the models are a classification problem rather than a regression: “A single neural network for binary classification for ICU outcome was developed for each future prediction time-period.” 

We have further emphasized that the neural networks are classifying care location in the third paragraph of the ‘Machine learning methodology’ section, first sentence: “The single-site CI-D models’ classification performance was first assessed in the test sample using the same performance metrics detailed above.” 

c) The authors declared that the objective of teach training epoch was to maximize the Mathew correlation coefficient and minimize the cross entropy between predicted score and the patient’s outcome, however, it’s still not clear what is the loss/cost function? Is it cross entropy only, or a joint loss of cross entropy and (reversed) correlation coefficient?

The methodology is detailed in referenced manuscripts (see references 8, 9, 10, 18) and only summarized in this one. To clarify the reviewer’s questions regarding model training: at every epoch, we attempted to minimize the cross entropy of the training set while monitoring the Mathew Correlation Coefficient (MCC) of the validation and the training sets. The numerical method we used to minimize cross entropy adds an additional step towards attaining a (possible local) minimum of the training cross entropy function. After this numerical step, we computed the MCC of both the training and validation sets. We strived to maximize the MCC of the validation set while monitoring the MCC of the training set. 

The training loss/cost function is a mixture of both the MCC and the cross entropy. The minimization algorithm works on the cross entropy; however it is accepted based on the behavior of the MCC of the validation and implicitly on its difference with the training MCC. After we achieved convergence (to a minimum), we assessed for overfitting and underfitting using the described additional derivations of confusion matrices on both the training and validation sets. The minimization of the training cross entropy, the maximization of validation MCC and its difference to the training MCC is a dynamic process. We minimized overfitting by using the derivations of confusion matrices at different cut points a posteriori. We assessed over fitting or underfitting by evaluating the trajectory plots of derivations of confusion matrices. We started the process of training again by using the weights up to the moment there was no overfitting/underfitting. The plot below is the after-the-fact check for overfitting/underfitting of a model trained with 240 epochs.

We have updated the maximization/minimization objective in the main document as follows: Machine Learning Methodology>Para 2: Amended sentence using the following sentence in paragraph 2 of methods: machine learning methodology:

“The objective of each training epoch was to minimize the training binary cross entropy while monitoring local maximization of the validation Mathew correlation coefficient (MCC) at a cut point of 0.5 while not allowing for drift from the training MCC at same cut point.”

d) What does the data look like. Is it time-series? Discrete numeric features? other type of features?

Note that this information is detailed in (and referenced to) previous manuscripts. The data is time series with both discrete numeric features and continuous values. Details of the dataset and elements included in the model are in the appendices 1-3 of the manuscript. For reference, here is a table from appendix 1:

Lab Variables1,3,4 Vital Signs1,3,4 Medications2,3 Other

Albumin Bilirubin Indirect Hemoglobin Platelets BP-systolic 1113 individual medications Age

Sex

ALT Bilirubin Total Hematocrit Potassium BP- diastolic 143 medication categories (6) 

Arterial Lactate BUN INR Protime Heart Rate 

PO2 (arterial) Calcium Glucose Sodium Respiratory Rate 

AST Calcium Ionized PTT Total Protein Temperature 

Base Excess Chloride PCO25 Venous Lactate Coma Score 

Bicarbonate Creatinine pH5 WBC 

Bilirubin Direct Fibrinogen 

1. Summarized for modeling with the following statistics for each variable: the count, sample mean, sample standard deviation (0 if the count was <2), maximum, and minimum. There were a total of 934 variables used for modeling: 300 derived from the 30 laboratory variables, 60 from the 6 vital signs, and 572 from the 143 medication categories, sex and age of patient at admission.

2. Summarized for modeling with the following statistics: the 6-hour sum per medication category of the number of medications given each hour; 2) the count of the previous time periods per medication category that the patient received one or more medications; 3) the proportion of the previous time periods per medication category that the patient received one or more medications.

3. Therapeutic intensity is reflected in the number of vital sign and laboratory measurements and medications.

4. If during the first six-hour time period there were missing values, these values were adjusted to the median of the first six-hour time periods adjusted to the following age groups: <1week, 1week-<4weeks, 4weeks-<3months, 3months-<1year, 1year-<2years, 2years-<3years, 3years-<8years, 8years-<12years, 12years-<22years.

5. Arterial, venous, capillary.

6. Classified by Multum1

e) What exactly is the structure of the neural network? The authors started with a neural network with a single hidden layer, and tried many things later, however, did not declare their decision of the network, nor did they show which attempt helped and which did not.

Again, this information is in the previous manuscripts and in the additional information provided above. The final structure of the neural network is explained in the response to the next question.

While it is not possible to explore all possible structures within the space of fully connected neural networks, we are confident that our final models are optimal for fully connected neural networks and less than 9 hidden layers. Some models with more than five hidden layers had similar performances to our final models, this was at the cost of large regularization parameters. We chose models that were as simple as possible while maintaining performances. The training process we described led us to the final models.

We explored different model spaces such as RNN, LSTM, GRU, CNN, achieving lower classification performances. It is not withing the scope of this paper to show these details.

f) What neural network model is applied? Fully-connected network? RNN/LSTM? CNN?

Note that this is not a methodology paper but an application manuscript for a medical journal. The goal is to alert readers for performance issues associated with machine learning models. Since these are beginning to be used in both medical research and practice, we believe this is a relevant topic. 

The second paragraph of the section “Machine learning Methodology, ” last sentence reads:

“The final neural networks were fully connected with five sequential hidden layers, but different number of nodes across hidden layers and models. Each layer had L1, L2 and dropout node regularizations with varying smoothing parameters. Each model has an output layer with one node and logistic activation.”

Also, second paragraph of the “Machine Learning Methodology” section reads: 

“If overfitting was found, we add L1 and L2 regularization, and layer node dropouts with parameters tuned to maintain similar metrices on the validation and training sets”

Third paragraph of the same section reads:

“If overfitting was found, we add L2 regularization, and layer node dropouts with parameters tuned to maintain similar metrices on the validation and training sets”

4. We previous suggested the authors to add other models to compare with: a). Traditional machine learning such as random forest, logistic regression. b). LSTM if time-series data. This is because neural networks do not always perform better than traditional machine learning models. Also, if the data is time-series, LSTM is a better neural network than fully-connected networks. 

As noted above, this is not a methodology paper but an application manuscript for a medical journal. The goal is to alert readers for performance issues associated with machine learning models. Since these are beginning to be used in both medical research and practice, we believe this is a relevant topic. 

For the reviewer’s reference, we did explore other modeling approaches prior to arriving at the current approach. As the reviewer suggested, we trained basic recurrent neural networks, gru, and lstm architectures, however these models had lower overall performance. To ensure that the model accounted for past clinical data in future predictions, we explicitly provide 24 hours of prior data for each patient as inputs in our neural networks. This methodology resulted in the best performing models among the different modeling methods we attempted.

5. The way to design the network is ad-hoc. The authors tried a few random things and search which works the best, however, is not an ideal way of designing neural networks. The design of a network needs reasons and domain knowledge support. Trying many things cannot tell readers any useful information when they want to apply the paper to their own work, and there are always new techniques to try.

As noted above, this is not a methodology paper but an application manuscript for a medical journal. The goal is to alert readers for performance issues associated with machine learning models. Since these are beginning to be used in both medical research and practice, we believe this is a relevant topic. The issue in the manuscript is not “how to design a neural network” but the performance of the network developed on what is often considered ”gold standard” data (multi-institutional data) in a new data set. The logic behind the evolution of this network is contained in the series of papers that preceded it. The focus on this paper is how the medical issues influenced the networks, more than the specifics of the networks. This is meant to be a manuscript for a medical audience.

---

## [Decision Letter · Decision Letter 2]

28 Apr 2023

PONE-D-22-13708R2External Evaluation of the Dynamic Criticality Index: A Machine Learning Model to Predict Future Need for ICU Care in Hospitalized Pediatric Patients.PLOS ONE

Dear Dr. Patel,

Thank you for submitting your manuscript to PLOS ONE. After careful consideration, we feel that it has merit but does not fully meet PLOS ONE’s publication criteria as it currently stands. Therefore, we invite you to submit a revised version of the manuscript that addresses the points raised during the review process.

There were some additional clarifications needed regarding modeling. Please address the additional reviewer concerns.==============================

We look forward to receiving your revised manuscript.

Kind regards,

Bobak J. Mortazavi, PhD

Academic Editor

PLOS ONE

Journal Requirements:

Reviewers' comments:

Reviewer's Responses to Questions

**Comments to the Author**

1. If the authors have adequately addressed your comments raised in a previous round of review and you feel that this manuscript is now acceptable for publication, you may indicate that here to bypass the “Comments to the Author” section, enter your conflict of interest statement in the “Confidential to Editor” section, and submit your "Accept" recommendation.

Reviewer #1: All comments have been addressed

Reviewer #3: All comments have been addressed

2. Is the manuscript technically sound, and do the data support the conclusions?

Reviewer #1: (No Response)

Reviewer #3: Yes

3. Has the statistical analysis been performed appropriately and rigorously? 

Reviewer #1: (No Response)

Reviewer #3: Yes

4. Have the authors made all data underlying the findings in their manuscript fully available?

Reviewer #1: (No Response)

Reviewer #3: Yes

5. Is the manuscript presented in an intelligible fashion and written in standard English?

Reviewer #1: (No Response)

Reviewer #3: Yes

6. Review Comments to the Author

Reviewer #1: (No Response)

Reviewer #3: Thanks for addressing my previous comments. They look good!

Some remaining comments are about section Machine learning methodology:

1. Line 210 "at cut points of 0.15, 0.5, and 0.9 in the training and validation sets": it's not very clear what are these cutting points about.

2. I don't think it's appropriate to say "if there was not overfitting". Overfitting will happen as the model is being trained, and it's important to catch the well-trained model before overfitting. E.g., using a validation and a test set - save the best performed model in validation and test it on test set.

3. Line 219 "with minibatches of 10,000": is it 10,000 batches or with batch size 10,000. Also, the paper of citation 19 uses minibatch of size ten.

7. PLOS authors have the option to publish the peer review history of their article (what does this mean?). If published, this will include your full peer review and any attached files.

Reviewer #1: No

Reviewer #3: **Yes: **Lida Zhang

---

## [Author Response · Author response to Decision Letter 2]

17 May 2023

General Comments: 

We are pleased that we seemed to have successfully responded to Reviewers 1 and 2 and hope that these responses to Reviewer 3’s final modelling questions appropriately address all remaining concerns.

1. Line 210 "at cut points of 0.15, 0.5, and 0.9 in the training and validation sets": it's not very clear what are these cutting points about.

Thank you for this clarifying question. We have now emphasized the definition of the cut points in relation to the output of the neural networks in following locations: (lines in red are new edits to the manuscript, whereas lines in black were already present but pasted below for context)

Line 202: “A single neural network for binary classification for ICU outcome was developed for each future prediction time-period. The raw output of the neural networks was a continuous number between 0 and 1 which was converted to a binary classifier (ICU care or non-ICU care) by choosing a cut point. A neural network output below the set cut point was classified as non-ICU care, whereas an output above the set cut point was classified as ICU care.”

Line 208: “We started with a neural network with a single hidden layer, and logit output and consecutively increased the number of nodes while monitoring for overfitting by computing the MCC, sensitivity, specificity, precision, negative predictive value at cut points of 0.15, 0.5, and 0.9 for binary classification in the training and validation sets.” 

2. I don't think it's appropriate to say "if there was not overfitting". Overfitting will happen as the model is being trained, and it's important to catch the well-trained model before overfitting. E.g., using a validation and a test set - save the best performed model in validation and test it on test set.

To address this concern, we have now amended this section as follows:

Line 208: “We started with a neural network with a single hidden layer, and logit output and consecutively increased the number of nodes while monitoring for overfitting by computing the MCC, sensitivity, specificity, precision, negative predictive value at cut points of 0.15, 0.5, and 0.9 for binary classification in the training and validation sets.” 

3. Line 219 "with minibatches of 10,000": is it 10,000 batches or with batch size 10,000. Also, the paper of citation 19 uses minibatch of size ten.

We have now amended Line 220 as follows: “An initial learning rate of 0.0002 was used with minibatches each consisting of a size of 10,000 data points.”

The reference that the reviewer is referring to regarding minibatches was provided as a reference for the technique we implemented in our model development, not the absolute value of the minibatch size.

---

## [Decision Letter · Decision Letter 3]

22 Jun 2023

External Evaluation of the Dynamic Criticality Index: A Machine Learning Model to Predict Future Need for ICU Care in Hospitalized Pediatric Patients.

PONE-D-22-13708R3

Dear Dr. Patel,

We’re pleased to inform you that your manuscript has been judged scientifically suitable for publication and will be formally accepted for publication once it meets all outstanding technical requirements.

Kind regards,

Bobak J. Mortazavi, PhD

Academic Editor

PLOS ONE

Additional Editor Comments (optional):

Please do consider the remaining comments from the reviewers for your final version.

Reviewers' comments:

Reviewer's Responses to Questions

**Comments to the Author**

1. If the authors have adequately addressed your comments raised in a previous round of review and you feel that this manuscript is now acceptable for publication, you may indicate that here to bypass the “Comments to the Author” section, enter your conflict of interest statement in the “Confidential to Editor” section, and submit your "Accept" recommendation.

Reviewer #3: All comments have been addressed

2. Is the manuscript technically sound, and do the data support the conclusions?

Reviewer #3: Yes

3. Has the statistical analysis been performed appropriately and rigorously? 

Reviewer #3: Yes

4. Have the authors made all data underlying the findings in their manuscript fully available?

Reviewer #3: Yes

5. Is the manuscript presented in an intelligible fashion and written in standard English?

Reviewer #3: Yes

6. Review Comments to the Author

Reviewer #3: A few remaining comments:

Limitation section: a) for the third point, there are techniques like SHAP which can provide interpretation of feature importance b) in addition to limitation, it will be great if the authors can also provide some information about future plan to address these limitation.

Line 230: please add citation to LIME "Ribeiro, Marco Tulio, Sameer Singh, and Carlos Guestrin. “” Why should i trust you?” Explaining the predictions of any classifier.” Proceedings of the 22nd ACM SIGKDD international conference on knowledge discovery and data mining. 2016."

Line 299-306: please be consistent - a) please give equation of Sensitivity (=true positives/(true positives+false negatives)), b) please be consistent about parentheses and square brackets in equations.

Line 409-416: Seem like the same content to Line 299-306.

7. PLOS authors have the option to publish the peer review history of their article (what does this mean?). If published, this will include your full peer review and any attached files.

Reviewer #3: **Yes: **Lida Zhang

---

## [Editor Report · Acceptance letter]

19 Jan 2024

PONE-D-22-13708R3 

PLOS ONE

Dear Dr. Patel, 

I'm pleased to inform you that your manuscript has been deemed suitable for publication in PLOS ONE. Congratulations! Your manuscript is now being handed over to our production team.

Kind regards, 

on behalf of

Dr. Bobak J. Mortazavi 

Academic Editor

PLOS ONE